# Supporting cells remove and replace sensory receptor hair cells in a balance organ of adult mice

Stephanie A Bucks[1], Brandon C Cox[2,3], Brittany A Vlosich[1], James P Manning[1], Tot B Nguyen[1], Jennifer S Stone[1]*

[1]Department of Otolaryngology-Head and Neck Surgery, Virginia Merrill Bloedel Hearing Research Center, University of Washington, Seattle, United States; [2]Department of Pharmacology, Southern Illinois University School of Medicine, Springfield, United States; [3]Department of Surgery, Division of Otolaryngology, Southern Illinois University School of Medicine, Springfield, United States

**Abstract** Vestibular hair cells in the inner ear encode head movements and mediate the sense of balance. These cells undergo cell death and replacement (turnover) throughout life in non-mammalian vertebrates. However, there is no definitive evidence that this process occurs in mammals. We used fate-mapping and other methods to demonstrate that utricular type II vestibular hair cells undergo turnover in adult mice under normal conditions. We found that supporting cells phagocytose both type I and II hair cells. *Plp1-CreER^T2*-expressing supporting cells replace type II hair cells. Type I hair cells are not restored by *Plp1-CreER^T2*-expressing supporting cells or by *Atoh1-CreER^TM*-expressing type II hair cells. Destruction of hair cells causes supporting cells to generate 6 times as many type II hair cells compared to normal conditions. These findings expand our understanding of sensorineural plasticity in adult vestibular organs and further elucidate the roles that supporting cells serve during homeostasis and after injury.

*For correspondence: stoner@uw.edu

**Competing interests:** The authors declare that no competing interests exist.

## Introduction

In the adult mammalian nervous system, production of sensory and neural cells is limited to a few areas, including the hippocampus, the olfactory epithelium, and the olfactory bulb (reviewed in *Beites et al., 2005*; *Ming and Song, 2005*). When neurons die naturally in these areas, their debris is phagocytosed by microglia (reviewed in *Fu et al., 2014*). Different glial populations then divide and generate replacement neurons (*Morshead et al., 1994*; *Doetsch et al., 1999*; *Johansson et al., 1999*).

In non-mammalian vertebrates, the mechanosensory receptors for balance, called vestibular hair cells (HCs), undergo turnover (cell death and replacement) throughout life. In the avian utricle, a vestibular organ sensing linear head movements, HCs die and are replaced at a slow rate in adulthood, maintaining cell numbers (*Jørgensen and Mathiesen, 1988*; *Roberson et al., 1992*; *Kil et al., 1997*; *Goodyear et al., 1999*). In mice, vestibular HC production is reported to occur only during gestation and the first two postnatal weeks (*Ruben, 1967*; *Rüsch et al., 1998*; *Kirkegaard and Nyengaard, 2005*; *Burns et al., 2012*). However, dying and immature-appearing HCs have been detected in utricles under normal conditions in adult guinea pigs (*Forge et al., 1993*; *Li et al., 1995*; *Rubel et al., 1995*; *Lambert et al., 1997*; *Forge et al., 1998*; *Forge and Li, 2000*) and bats (*Kirkegaard and Jørgensen, 2000, 2001*). The ability of adult rodents to regenerate small numbers of utricular HCs after ototoxin-induced damage is another indicator of plasticity in mammalian vestibular epithelia (*Forge et al., 1993*; *Warchol et al., 1993*; *Forge et al., 1998*; *Kawamoto et al.,*

**eLife digest** Cells in the inner ear called hair cells sense sound waves and head movements, allowing us to hear and maintain balance. In non-mammals such as birds and fish, the hair cells responsible for balance die and are replaced (in a process known as turnover) throughout life. However, it is largely assumed that no new balance hair cells are made in adult mammals such as humans and mice. This would mean that injured hair cells are never replaced, which could cause balance problems such as dizziness over time.

There have been hints in past studies that perhaps some balance hair cells die or are newly made in adult mammals. Using a variety of new cell labeling and tracking methods in different types of mutant mice, Bucks et al. now show that the turnover of balance hair cells happens in adult mice under normal conditions. Both types of balance hair cells – known as type I and type II – are removed by supporting cells that surround the hair cells. In addition, the supporting cells can convert into new type II hair cells, but not type I hair cells, and type II hair cells do not convert into new type I hair cells.

To compare these results with what happens after hair cell damage, Bucks et al. injected a toxin into mutant mice to kill most hair cells. This revealed that supporting cells make 6 times as many hair cells after severe damage than under normal conditions, but still only make type II hair cells.

One important issue to study next is whether type I hair cells are ever created in adulthood. Many elderly people develop balance problems that lead to catastrophic falls. Perhaps one reason this occurs is because type I hair cells cannot be replaced in humans.

*2009*; *Golub et al., 2012*). Many tissues capable of regeneration after injury, such as integumentary, olfactory, and intestinal epithelia, also undergo cellular turnover under normal conditions (*Taylor et al., 2000*; *Ito et al., 2005*; *Barker et al., 2007*; *Leung et al., 2007*). Collectively, these studies suggest vestibular HC turnover may occur in adult mammals, but definitive evidence has not been presented.

Supporting cells (SCs), which surround HCs, are epithelial cells that have properties of glia. Among their many functions (reviewed in *Monzack and Cunningham, 2013*; *Wan et al., 2013*), SCs serve as progenitors to new vestibular HCs during homeostasis in mature birds (*Jørgensen and Mathiesen, 1988*; *Roberson et al., 1992*; *Kil et al., 1997*) and after ototoxin-induced death of HCs in mature birds and mammals (*Tsue et al., 1994*; *Lin et al., 2011*). SCs also act as phagocytes after ototoxic damage, clearing cellular debris in birds (*Bird et al., 2010*) and mammals (*Monzack et al., 2015*). The capacity of SCs in adult mammals to phagocytose and renew vestibular HCs under normal conditions has not been explored.

In this study, we demonstrate that a small, but significant proportion of HCs in utricles of adult mice are cleared and replaced by SCs in the absence of a damaging stimulus. There are two types of vestibular HCs, type I and type II, both of which are phagocytosed and removed from the sensory epithelium. However, fate-mapping indicates that SCs replace only type II HCs, and type II HCs do not convert into type I HCs, at least over the 8-month period of adulthood we examined. When HCs are killed by a toxin, SCs transdifferentiate into 6 times as many type II HCs compared to normal HC addition within the same 4-week period. This study demonstrates that the utricle is an additional neuroepithelium in mammals capable of generating sensory receptor cells throughout adulthood and defines the lineage of new HC production in this mature tissue under normal conditions and after HC damage *in vivo*.

## Results

### Type I and II HCs are cleared by phagosomes from adult mouse utricles under normal conditions

Utricles are otolithic vestibular organs located in the inner ear (*Figure 1A*). The utricular sensory epithelium (macula) is composed of an alternating array of SCs and two types of HCs (*Figure 1B–D*) and contains the processes of vestibular afferent and efferent nerves (reviewed in *Eatock and*

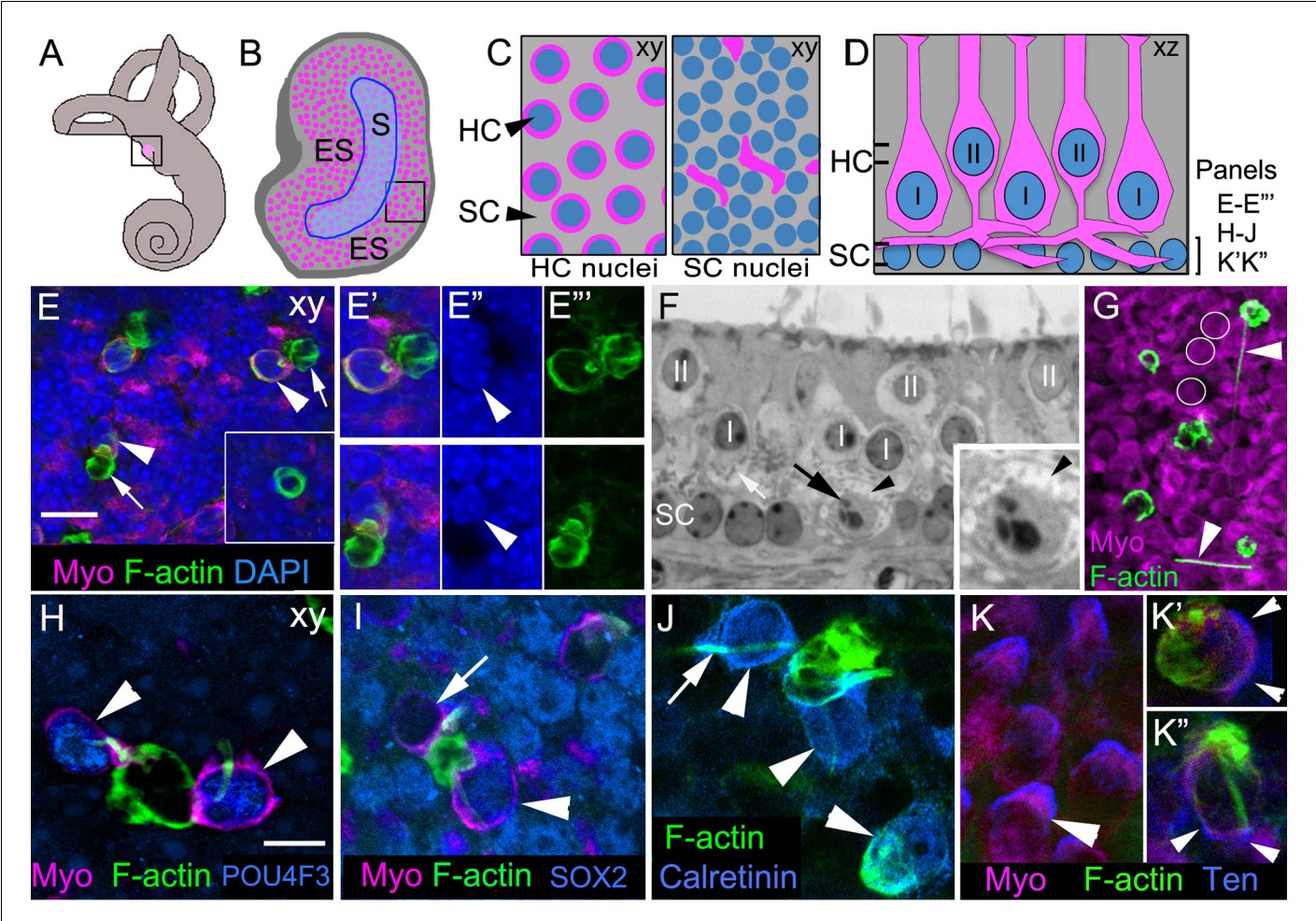

**Figure 1.** Phagosomes target type I and type II HCs for clearance in adult mouse utricular maculae under normal conditions. (**A**) Schematic of the inner ear with the utricle highlighted in magenta. (**B**) Schematic of a surface view of a utricle (xy view) with HCs in magenta. Blue outlined region denotes the striola (S). The region surrounding the striola is the extrastriola (ES). Boxed area shown at higher magnification in **C**. (**C**) Higher magnification of schematic in **B** with slices through the level of type II HC nuclei (left panel) and the level of the SC nuclei (right panel). Nuclei in blue, HC perinuclear cytoplasm (left panel) and basolateral processes (right panel) in magenta, and SC cytoplasm in grey (xy view). (**D**) Schematic of a cross-section (xz view) of the adult mouse utricle. II, type II HCs; I, type I HCs. Double lines indicate positions of the xy views shown in **C**, with the upper set of lines referring the left panel in **C**, and the lower set of lines referring to the right panel in **C**. The bracket indicates the level of the SC nuclei, the focal plane of xy confocal optical images in **E–E'''**, **H–J**, **K'–K''**. In panels **E–K''**, F-actin was labeled with phalloidin (green) and HCs were labeled with anti-myosin VIIa antibodies (Myo, magenta), except there is no myosin label in panel **J**. In **E–E'''**, blue label is DAPI. In **H–K''**, each blue label is a different cell-selective marker. (**E–E'''**) Confocal xy optical sections of the SC nuclear layer in an adult Swiss Webster utricle. (**E**) Two ectopic HCs (arrowheads) are located next to F-actin-rich phagosomes (arrows). Inset, a ring-shaped phagosome not associated with a HC. See *Video 1* for a 3D reconstruction of a phagosome targeting a HC, and see *Video 2* for all xy images in the z-series of **E**. Myosin-labeled HC cytoplasm that is not surrounding a nucleus corresponds to type II HC basolateral processes (see 1C, right panel). (**E'–E''**) Higher magnification of the two HCs indicated by arrowheads in **E**. Arrowheads in **E''** point to the nucleus of each ectopic HC. (**F**) Transverse section of an adult Swiss Webster mouse utricle showing an ectopic HC (black arrow) located in the SC nuclear layer that has condensed chromatin. The ectopic cell is surrounded by a calyx (black arrowhead), typical of a type I HC. The calyx appears as an electron-lucent ring around the cell, which has minimal cytoplasm and contains numerous electron-dense mitochondria (small gray dots). A calyx surrounding a normally localized type I HC (I) is indicated by the white arrow. Several normally positioned type I and II HCs are indicated (I, II over nucleus). Inset, higher magnification of the ectopic type I HC. (**G**) Several HCs and F-actin-rich phagosomes are shown in this projection image of a Swiss Webster macula. Two F-actin spikes are indicated by arrowheads. White circles indicate the area of 3 type II HCs for reference. (**H–K''**) Phagosomes co-localized with markers of type I or II HCs in xy confocal slices at the level of SC nuclei in Swiss Webster utricles. (**H**) Two ectopic HCs (arrowheads) have POU4F3-positive nuclei (blue) and are connected to a large ring-shaped phagosome. (**I**) Two ectopic HCs associated with a basket-like phagosome. One HC has a SOX2-negative nucleus (arrow, lacking blue label), and the other HC has a SOX2-positive nucleus (arrowhead, blue). (**J**) Three ectopic HCs (arrowheads) associated with phagosomes are calretinin-positive (blue). Arrow indicates an example of a F-actin spike. (**K**) Tenascin (Ten, blue) immunolabeling in normally localized HCs (arrowhead). (**K',K''**) Tenascin labeling (blue) is evident in two ectopic HCs (arrowheads) co-

*Figure 1 continued on next page*

*Figure 1 continued*

localized with phagosomes (green). Scale bar shown in **E** is 10 µm for **E**, 7 µm for **E'–F**, 3.5 µm for **F** inset, and 14 µm for **G**. Scale bar shown in **H** is 5 µm for **H–K''**.

*Songer, 2011*). Type I HCs have a flask-shaped body and a nucleus located at mid-epithelial depth, and they synapse onto large calyceal afferents. Type II HCs have complex shapes, with thick necks and basal cytoplasmic processes projecting from the cell body (*Figure 1C,D*; *Pujol et al., 2014*). Each type II HC nucleus is located near the lumenal surface. Type II HCs synapse onto small (bouton) afferents. SCs span the basal-to-apical extent of the epithelium, and their nuclei reside below HC nuclei, near the basal lamina. The utricle is divided into two zones: a C-shaped central region called the striola, in which specialized type I HCs are enriched, and the extrastriolar region, which surrounds the striola and occupies the remainder of the utricle (*Figure 1B*).

To address if vestibular HCs undergo turnover in normal adult mice, we sought evidence for programmed cell death in whole-mounted utricles utilizing terminal deoxynucleotidyl transferase dUTP nick end labeling (TUNEL) and immunolabeling for activated caspase-3 (aCasp3). Although we detected cells with TUNEL or aCasp3 immunolabeling in connective tissue below the macula and in intestinal epithelia labeled simultaneously with utricles, we detected no labeled cells in normal maculae using either method (data not shown).

We reasoned that HC removal may occur too infrequently to capture cell death, so we looked for other evidence. When SCs clear HCs after ototoxin-mediated injury, they first produce an actin cable to constrict the HC and extrude its most apical portion, including the stereocilia bundle (*Meiteles and Raphael, 1994*; *Li et al., 1995*; *Bird et al., 2010*; *Monzack et al., 2015*). Then, one or more SCs form actin-rich phagosomes that engulf the remaining HC body. Actin filaments comprising the phagosome are interlaced around each HC as it is consumed, and the phagosome structure appears basket- or ring-like (*Bird et al., 2010*; *Monzack et al., 2015*).

To determine if utricular SCs consume HCs under normal conditions in adult mice, we

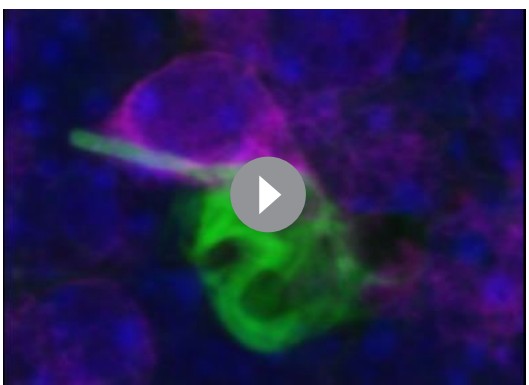

**Video 1.** 3D reconstruction of a phagosome in an adult Swiss Webster utricle under normal conditions. This movie is a y-axis rotation of a 3D reconstruction of an ectopic HC (myosin VIIa, magenta) associated with a F-actin phagosome (phalloidin, green). The phagosome consists of a large basket-like structure with a spike, or process, that extends through the cytoplasm of the HC along its nucleus. Note the basket-like structure is not solid F-actin but has gaps that lack labeling suggestive of a lattice. All nuclei are labeled with DAPI (blue).

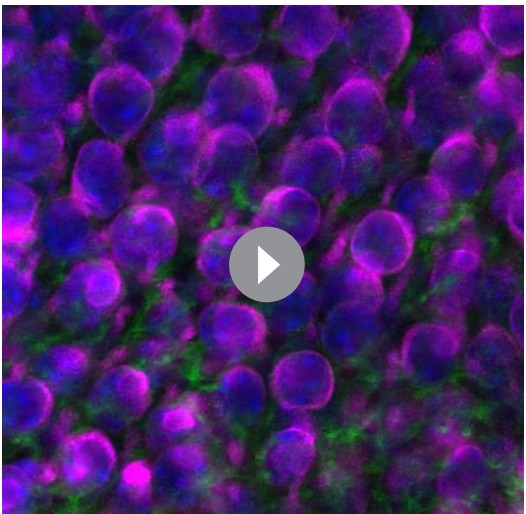

**Video 2.** Phagosomes in the SC nuclear layer of an adult Swiss Webster utricle under normal conditions. This movie was constructed from optical sections collected from the area shown in *Figure 1E*. It begins at the level of type II HC nuclei and progresses through type I HC and SC nuclei, ending at the basal lamina. All nuclei are labeled with DAPI (blue), HCs are labeled with myosin VIIa antibodies (magenta), and the F-actin in phagosome basket- and ring-like structures and spikes (or processes) is labeled with phalloidin (green).

labeled utricles from 5- to 10-week-old Swiss Webster mice with phalloidin, which binds filamentous actin (F-actin). In whole-mounted normal utricles, we detected numerous F-actin-rich structures that were basket-like or ring-shaped (*Figure 1E–E'''',G–K''*, *Video 1*), resembling phagosomes described before (*Bird et al., 2010*; *Monzack et al., 2015*). The vast majority of these structures was restricted to the basal compartment of the epithelium, amongst SC nuclei. We counted 48.9 ± 14.8 (mean ± standard deviation) phagosomes per utricle (n = 8; Figure 3A; Figure 3—source data 1). Immunolabeling of myosin VIIa, a HC-specific marker (*Hasson et al., 1995*), revealed that 21.7% ± 16.1% of phagosomes per utricle were clearly associated with HCs whose cell bodies had been translocated basally to the SC nuclear layer and were therefore considered ectopic (95% confidence interval: 8.8–34.6%; n = 6; *Figure 1E–E''',H–K''*; *Video 2*). There were two primary types of associations between phagosomes and ectopic HCs. In 55.2% (±9.7%; 95% confidence interval: 47.5–63.0%; n = 6) of associations, phagosomes were composed of a ring-like structure that fully encircled a HC body (*Figure 1E–E'''*, arrowheads) and were connected to a basket-like structure devoid of HC material (*Figure 1E–E'''*, arrows). Most other phagosomes associated with ectopic HCs consisted of a basket-like structure with one or more F-actin-rich processes that extended laterally and either contacted nearby HCs (*Figure 1G*) or pierced and entered their cytoplasm (*Figure 1H,J,K''*, *Video 1*). In Swiss Webster mice, we counted 12 ± 9.9 ectopic HCs per utricle that were associated with phagosomes (95% confidence interval: 4.3–20.1; n = 6). However, the majority of phagosomes (71.7% ± 14.3%; 95% confidence interval: 60.3–83.1%; n = 6) lacked HC staining within or around them (*Figure 1E* inset), suggesting they may be clearing other cell types or being retracted after completion of HC digestion.

Examination of serial transverse sections of a utricle from an adult Swiss Webster mouse confirmed that some HCs were located in a basal, ectopic position (*Figure 1F*). Of 6 such cells we observed in the sectioned utricle, one HC appeared to have condensed chromatin characteristic of apoptosis (*Figure 1F*). Although we examined hundreds of ectopic HCs in utricles, no other HCs had obvious abnormalities or signs of apoptosis other than their location. Indeed, 4',6-diamidino-2-phenylindole (DAPI) labeling revealed no variation in nuclear size, integrity, or density of chromatin in ectopically located HCs or HCs associated with phagosomes (*Figure 1E''*).

To examine the type of HCs targeted by phagosomes, we labeled utricles with phalloidin and antibodies to other HC markers (*Figure 1H–K''*). The nuclei of basally translocated cells being engulfed by phagosomes were immunoreactive for antibodies against POU4F3 (*Figure 1H*), a HC-specific transcription factor (*Erkman et al., 1996*; *Xiang et al., 1997*), confirming their identity. We found evidence that both type I and type II HCs were being targeted by phagosomes. SOX2 is a transcription factor that is abundant in SCs and type II HCs, but is not found in type I HCs (*Oesterle et al., 2008*). Some phagosomes were associated with a myosin VIIa-positive cell whose nucleus was SOX2-positive (*Figure 1I*). In other utricles, we found phagosome-associated HCs that were immunoreactive for the calcium-binding protein, calretinin (*Figure 1J*), which is selectively elevated in type II HCs (*Desai et al., 2005*). Some phagosomes engulfed cells that were immunoreactive for tenascin (*Figure 1K–K''*), an extracellular matrix protein that lines the space between the type I HC plasma membrane and the afferent calyx (*Swartz and Santi, 1999*). Further, one ectopic HC in transverse plastic sections was wrapped by a calyceal afferent (*Figure 1F*), identifiable by its electron-lucent appearance except for scattered mitochondria, which is typical of a type I HC.

To address the mechanism of HC clearance, we examined phalloidin labeling in HCs located in their proper positions. We estimate that 2–10 normal-appearing HCs per utricle contained a long 'spike' of F-actin that coursed through the cytoplasm. Typically, the spike extended from the cell's apex to its base, then exited the cell and descended toward the basal lamina (*Figure 2A–A''*, *Video 3*). It is notable that we detected a similar spike-like structure in the cytoplasm of 5.3 (±4.3; 95% confidence interval: 2.0–8.7; n = 6) basally translocated HCs per utricle that were associated with phagosomes (*Figure 2B–B''*, and *Figure 1H,J,K''*). Some actin spikes extended laterally outside the HCs for several cell widths (*Figure 1G*), and some spikes were not clearly associated with any HCs.

During phagocytosis after HC damage by aminoglycosides, the apical portion of the HC is cleaved and ejected apically (*Meiteles and Raphael, 1994*; *Li et al., 1995*; *Bird et al., 2010*; *Monzack et al., 2015*). To determine if this also occurs under normal conditions, we labeled utricles with antibodies to espin and plasma membrane calcium ATPase 2 (PMCA2), which are reported to label stereocilia, but not other structures in the sensory epithelium (*Dumont et al., 2001*; *Li et al.,*

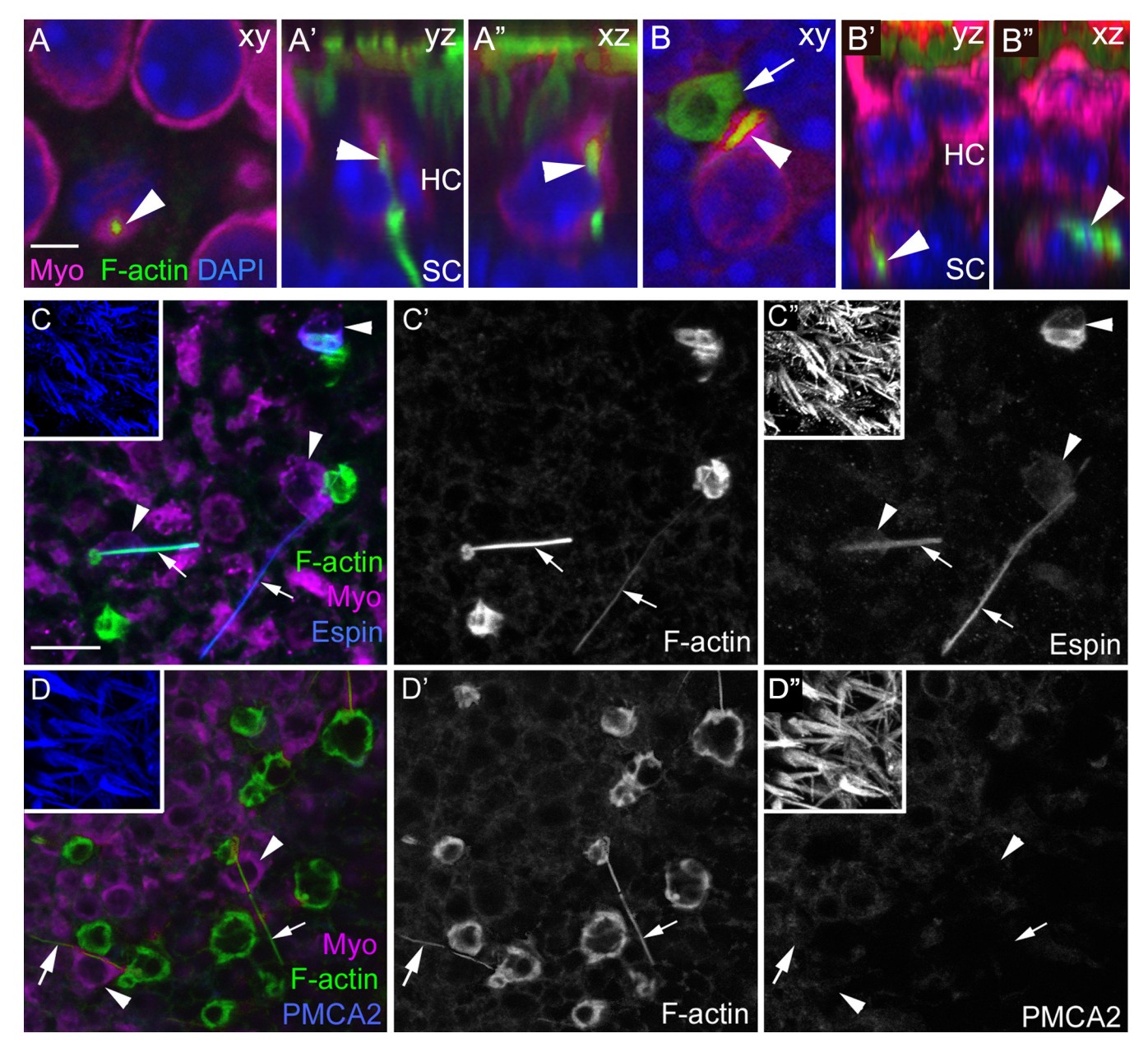

**Figure 2.** F-actin spikes are present in HCs, and the HC bundle is likely ejected prior to HC body translocation under normal conditions. (A–A") A F-actin spike (green, arrowhead) in a normally localized HC [myosin VIIa (Myo) magenta] from a C57Bl/6J mouse is shown in 3 views. (A) is a xy optical section at the level of type II HC nuclei (DAPI, blue). (A',A") are yz and xz optical (cross) sections of the HC indicated in A by the arrowhead. Note the F-actin spike is beneath the apical surface, which is highlighted by the row of brightly labeled stereocilia (green) at the top of A' and A". See *Video 3* for all xy images in the z-series of A. HC, HC nuclear layer; SC, SC nuclear layer. (B–B") A F-actin spike (green, arrowhead) in an ectopic HC (Myo, magenta) from a C57Bl/6J mouse is shown in 3 views. (B) is a xy optical section at the level of SC nuclei (DAPI, blue). (B',B") are yz and xz optical sections of the HC shown in B. Arrow points to a basket-like phagosome. (C–D'') Confocal optical sections from two normal Swiss Webster mouse utricles labeled with F-actin, antibodies to myosin VIIa, and with either anti-espin antibodies (C–C") or anti-PMCA2 antibodies (D–D"). The 3 panels for each utricle (C–C" or D-D") show different label combinations for the same field, as indicated. All panels are focused on the SC layer, except the boxed insets, which are focused on stereocilia. Arrowheads in C,D and C",D" point to ectopic HCs (Myo, magenta in C and D), while arrows in all panels point to F-actin spikes (green in C,D and white in C',D'). Scale bar in A is 3 µm and applies to A–B". Scale bar in C is 10 µm and applies to C–D", including insets.

*2004*). As anticipated, stereocilia were brightly labeled with both antibodies (*Figure 2C,D* insets). Occasional ectopic HCs were diffusely labeled by either antibody (*Figure 2C,C''*, espin labeling), but none of them contained brightly labeled foci resembling stereocilia (*Figure 2C–D''*). Thus, our observations suggest that the apical part of the HC is ejected apically prior to translocation or that stereocilia degenerate during translocation. Interestingly, espin antibodies labeled most actin-rich spikes (*Figure 2C–C''*). Initially, we thought this could indicate that spikes are derived from stereocilia. However, spikes were not labeled for antibodies to PMCA2 (*Figure 2D–D''*), suggesting this is not the case. Perhaps espin serves as an actin-bundling protein in phagosome spikes, as it does in stereocilia (*Zheng et al., 2000*).

In 5- to 10-week-old Swiss Webster mice, there were 48.9 (±14.8) phalloidin-labeled phagosomes per utricle (*Figure 3A*, *Figure 3—source data 1*), displaying a range of morphologies. To assess if phagosomes are unique to Swiss Webster mice at 5–10 weeks of age, we analyzed Swiss Webster utricles at 3 weeks and 43–46 weeks of age, as well as utricles from two strains of inbred mice (C57Bl/6J and CBA/CaJ) at all 3 ages (*Figure 3A*, *Figure 3—source 1*). The distribution of phagosomes in the utricular macula was similar across utricles and ages: phagosomes appeared to be concentrated in the peristriolar region (*Figure 3B,B'*). The morphologies of phagosomes were also similar in all 3 strains of mice (not shown). However, C57Bl/6J and CBA/CaJ utricles consistently had fewer phagosomes than Swiss Webster utricles (*Figure 3A*, *Figure 3—source data 1*). Within each strain, the number of phagosomes was similar at all ages examined (*Figure 3A*, *Figure 3—source data 1*). These findings demonstrate that HC clearance occurs under normal conditions in different strains of mice at a range of ages.

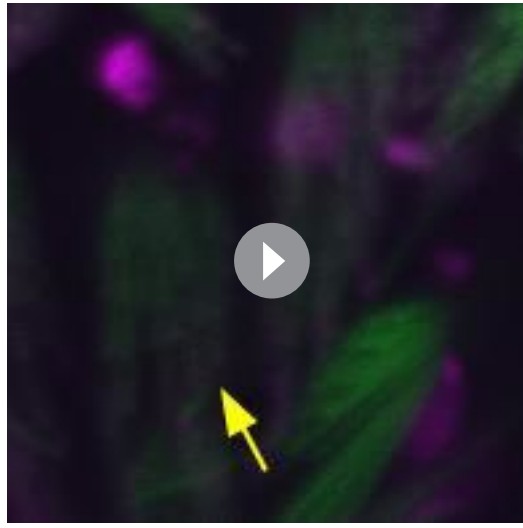

**Video 3.** F-actin spike through a normally located HC in an adult mouse utricle under normal conditions. This movie was constructed from optical sections collected from the area shown in *Figure 2A–A''*. The movie begins above the stereocilia bundles (F-actin, green), continues through the type II and type I HC nuclear layers (DAPI, blue), and ends in the SC nuclear layer. A myosin VIIa labeled HC (magenta), which has a normal-appearing stereocilia bundle (below the arrow), has a F-actin spike (green, arrow) within its cytoplasm, extending the length of the HC and into the SC nuclear layer.

## SCs clear HCs from adult mouse utricles under normal conditions

We examined whether phagosomes derive from SCs or from cells in the macrophage/monocyte lineage. First, we fluorescently labeled the cytoplasm of SCs to assess if they generate actin-rich phagosomes using *Plp1-CreER$^{T2}$* mice, which have been used previously to label SCs in mouse utricles (*Gómez-Casati et al., 2010*; *Burns et al., 2012*; *Wang et al., 2015*). In 6-week-old *Plp1-CreER$^{T2}$: ROSA26$^{CAG-loxP-stop-loxP-tdTomato}$* mice (hereafter referred to as *Plp1-CreER$^{T2}$:ROSA26$^{tdTomato}$* mice), the majority of SCs were tdTomato-positive at one week after injection of tamoxifen (*Figure 4B*). A small number of cells in the transitional epithelium, which borders the sensory epithelium (*Figure 4B*), and numerous cells in the stroma (presumed Schwann cells, not shown) were also tdTomato-positive. We sampled 8 regions of the macula and determined that 91.7% (±6.1%; n = 3) and 68.4% (±1.8%; n = 3) of SCs in the extrastiola and the striola, respectively, were tdTomato-positive (*Figure 4—source data 1*). In age-matched *Plp1-CreER$^{T2}$:ROSA26$^{tdTomato}$* mice that did not receive tamoxifen, <5% of SCs per utricle (126.8 ± 46.8; 95% confidence interval: 80.9–172.6; n = 4) were tdTomato-positive (*Figure 4A*), revealing some tamoxifen-independent Cre activity. We labeled *Plp1-CreER$^{T2}$:ROSA26$^{tdTomato}$* utricles collected at one week post tamoxifen with phalloidin to visualize phagosomes and antibodies against myosin VIIa to visualize HCs. We detected an average of 27.8 (±4.3; 95% confidence interval: 23.0–32.6; n = 3) phagosomes per utricle, which were fewer than Swiss Webster mice, but more than CBA/CaJ and C57Bl/6J mice (*Figure 3A*, *Figure 3—source*

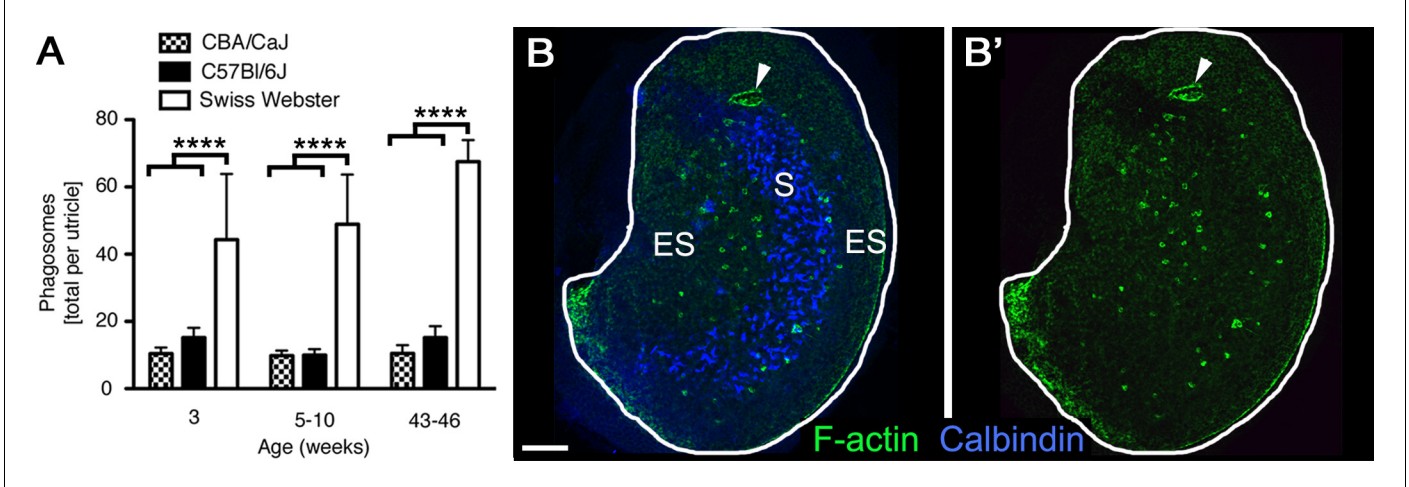

**Figure 3.** Phagosomes are present in several mouse strains across ages. (**A**) Number of phagosomes per utricle in 3 mouse strains (CBA/CaJ, C57Bl/6J, and Swiss Webster) at 3 weeks, 5–10 weeks, and 43–46 weeks of age. Data are presented as mean ± 1 standard deviation for n = 3–8 mice per group (see *Figure 3—source data 1*). Within each strain, the number of phagosomes did not increase significantly over time. CBA/CaJ and C57Bl/6J mice had similar phagosome numbers (10–15 per utricle), but Swiss Webster mice had significantly more phagosomes per utricle (45–65) than the other strains at every age, as determined by two-way ANOVA (p=0.0189 for age and p<0.0001 for strain followed by Bonferroni's multiple comparisons test; ****p<0.0001). (**B,B'**) Confocal projection image of the utricular macula from an adult Swiss Webster mouse showing that F-actin-rich phagosomes (green) are concentrated in the peristriolar region of the extrastriola (ES) but are largely absent from the calbindin-labeled striola (S, blue) and the peripheral-most portion of the ES. The projection image was constructed from just beneath the stereocilia through the SC layer to avoid obstruction of the phagosomes by the F-actin-rich stereocilia. Scale bar is 100 μm. Arrowheads in **B,B'** point to a portion of the utricular macula that sustained damage during dissection; this region is rich in F-actin but is not a phagosome.

The following source data is available for figure 3:

**Source data 1.** Quantification of phagosomes in the normal utricle of 3 mouse strains at 3 ages.

*data 1*) and 4.5 (±2.3; 95% confidence interval: 1.9–7.1; n = 3) phagosomes were associated with a HC. In some utricles, we detected overlap of tdTomato and phalloidin signals, indicating that some phagosomes were derived from SCs (*Figure 4C–E'*). It was unclear if phagosomes were generated by a single SC or by two or more adjacent SCs, but phagosomes were consistently derived from the most basal region of a SC (*Figure 4D–E'*).

SCs, cells in the transitional epithelium, and cells in the connective tissue underlying the sensory epithelium (presumed Schwann cells) were labeled by tdTomato in *Plp1-CreER^{T2}:ROSA26^{tdTomato}* mice. To further delineate which cell types contribute to the production of phagosomes, we examined utricles from *Lfng-eGFP* mice, which express eGFP under control of *Lfng* regulatory elements (*Gong et al., 2003*). In *Lfng-eGFP* utricles, eGFP was expressed in the majority of SCs, but not in HCs, cells in the transitional epithelium, or the underlying connective tissue (*Figure 4H–I'''*; *Burns et al., 2015*). F-actin-labeled phagosomes were also detected in the basal compartment of the sensory epithelium of *Lfng-eGFP* utricles, and some were co-labeled with eGFP (*Figure 4I–I''*), providing further evidence that phagosomes derive from SCs.

To assess if cells from the macrophage/monocyte lineage also phagocytose HCs under normal conditions, we labeled adult Swiss Webster utricles with myosin VIIa to detect HCs and ionized calcium-binding adaptor molecule 1 (IBA1) or CD68 to detect resting or activated macrophages, respectively (*Holness et al., 1993*; *Imai et al., 1996*; *Ito et al., 1998*). We also applied phalloidin to label phagosomes and DAPI to label nuclei. F-actin-labeled phagosomes in the sensory epithelium never co-localized with IBA1 labeling (*Figure 4F,G*) or CD68 labeling (not shown), although we consistently detected cells that were IBA1-positive (*Figure 4F',G*) or CD68-positive (not shown) in the connective tissue below the macula. These results are consistent with a previous study that did not detect macrophages in the utricular macula of normal adult mice (*Kaur et al., 2015*).

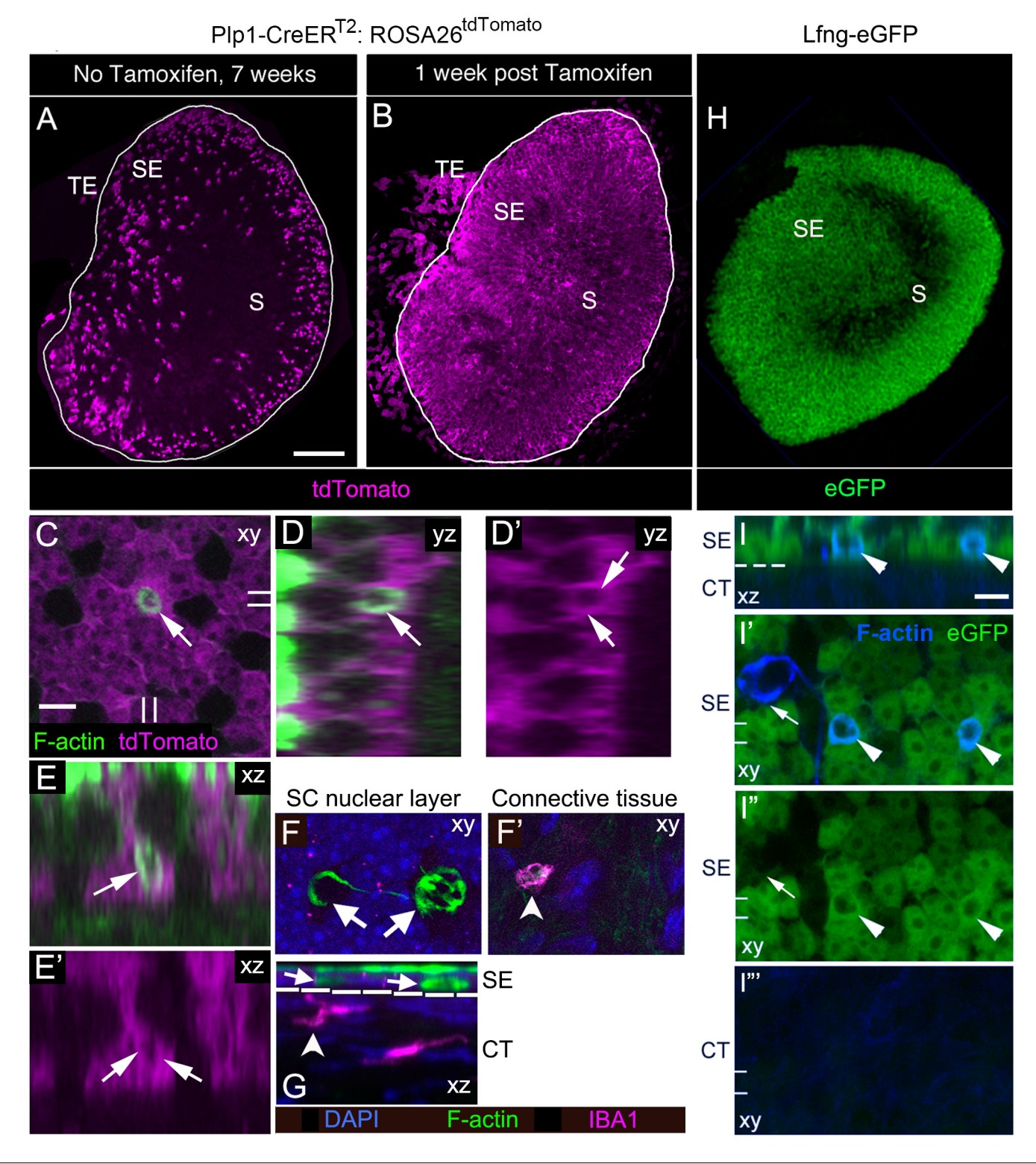

**Figure 4.** SCs, not macrophages, produce phagosomes in adult mouse utricles under normal conditions. (A,B) Confocal projection images of the utricular macula from 7-week-old *Plp1-CreER^T2:ROSA26^tdTomato* mice showing the numbers and distribution of tdTomato-positive SCs (magenta) in the utricular sensory epithelium of mice that received no tamoxifen (A) or mice that received tamoxifen (B). S, striola; SE, sensory epithelium; TE, transitional epithelium. See *Figure 4—source data 1* for quantification of tdTomato-positive SCs in extrastriolar and striolar regions. (C–E') Higher magnification optical sections of a *Plp1-CreER^T2:ROSA26^tdTomato* mouse utricular macula at one week post tamoxifen showing overlap between tdTomato-labeled SC

*Figure 4 continued on next page*

*Figure 4 continued*

cytoplasm (magenta) and a F-actin-rich phagosome (green, arrows). (**C**) Xy view with double lines indicating where cross-sectional images were created in **D-E'**. (**D,D'**) Yz view of same area in **C**.( **E,E'**) Xz view of same area in **C**. Very bright green labeling in **D** and **E** is F-actin in the stereocilia bundles of HCs. (**F,G**) Confocal optical sections of an adult Swiss Webster utricle. (**F**) Two F-actin-rich phagosomes (green, arrows) in the SC nuclear layer (DAPI, blue) did not co-label for antibodies to IBA1, a macrophage/monocyte lineage marker (magenta). (**F'**) An IBA1-positive cell (magenta, arrowhead) resided in the connective tissue under the phagosomes shown in **F**. (**G**) Xz view of the field shown in **F,F'**. The white dotted line indicates the border between sensory epithelium (SE) and connective tissue (CT). Arrows and arrowhead indicate the same cells shown in **F,F'**. (**H**) Confocal projection image of the utricular macula from an adult *Lfng-eGFP* mouse showing eGFP-positive SCs (green) in the sensory epithelium (SE). S, Striola. (**I–I'''**) Confocal optical sections of the same field from the extrastriolar region of a *Lfng-eGFP* utricle, with eGFP in green and F-actin in blue. (**I**) Xz view through the SE and CT of the utricle. (**I',I''**) Xy views of the SC layer in the SE. (**I'''**) Xy view of the CT. The dotted line in **I** is the approximate location of the basal lamina, between the SE and the CT. Two phagosomes (arrowheads, F-actin, blue) are flanked by eGFP-positive SCs (green) and are also co-labeled with eGFP (green); co-labeled phagosomes appear cyan. One phagosome (arrow in **I',I''**) is eGFP-negative and not flanked by eGFP-positive SCs. The double lines in **I–I'''** indicate the position of the xz view shown in **I**. Scale bar in **A** is 100 μm and applies to **A,B,H**. Scale bar in **C** is 5 μm and applies to **C–G**. Scale bar in **I** is 5 μm and applies to **I–I'''**.

The following source data is available for figure 4:

**Source data 1.** Quantification of the percentage of SCs labeled with tdTomato in *Plp1-CreER^{T2}:ROSA26^{tdTomato}* utricles.

Our observations demonstrate that SCs, not macrophages, clear type I and II vestibular HCs from adult mouse utricles under normal conditions by creating F-actin-rich phagosomes.

## Immature HCs are present in utricles of adult mice under normal conditions

Numbers of total utricular HCs and SCs seem to be stable in adult mice at the ages we examined (*Kirkegaard and Nyengaard, 2005*; *Burns et al., 2012*). Therefore we reasoned that if HCs are cleared from the macula, they must be replaced. To address this, we looked for evidence of ongoing HC addition by assessing if HCs in normal adult utricles express two markers specific for immature HCs, protocadherin15-CD2 (PCDH15-CD2) and ATOH1.

PCDH15 is a protein localized to the tip links of HC stereocilia. Two alternatively spliced variants of PCDH15 (CD1 and CD3) are present in mature stereocilia (*Ahmed et al., 2006*). In contrast, PCDH15-CD2 is detected along the length of stereocilia during early development, but becomes confined to the kinocilium once HCs mature (*Ahmed et al., 2006*; *Webb et al., 2011*). Thus, PCDH15-CD2 is a selective marker of immature stereocilia. As positive controls, we labeled utricles from neonatal (<8 day-old) Swiss Webster mice with antibodies to PCDH15-CD2 and with phalloidin to mark stereocilia. In the periphery of the utricle, where newly formed HCs are differentiating in neonates (*Burns et al., 2012*), we detected numerous HCs with PCDH15-CD2 labeling throughout the stereocilia (*Figure 5A*). Next, we examined utricles from 6- to 9-week-old CBA/CaJ and Swiss Webster mice. We observed small numbers of HCs with PCDH15-CD2 labeling throughout stereocilia in adult utricles (*Figure 5B*), primarily in the most peripheral portion of the extrastriolar region (*Figure 5C*). CBA/CaJ and Swiss Webster mice had 17.2 (±4.0) and 24.0 (±8.8) HCs with the immature labeling pattern for PCDH15-CD2, respectively (*Figure 5D*, *Figure 5—source data 1*).

ATOH1 is a transcription factor that is abundant in early differentiating HCs and weakly expressed in mature HCs (*Lanford et al., 2000*; *Chen et al., 2002*; *Woods et al., 2004*; *Cai et al., 2013*). We examined utricles collected from 6-week-old *Atoh1^{GFP/GFP}* mice, which express an ATOH1-GFP fusion protein under control of the endogenous *Atoh1* promoter (*Rose et al., 2009*). In *Atoh1^{GFP/GFP}* mice, ATOH1-GFP is abundant in developing cochlear HCs but is lost once HCs mature (*Cai et al., 2013*). Using antibodies to amplify the GFP signal, we detected numerous GFP-positive cells that were concentrated in the extrastriola (*Figure 5E*). We counted 82.3 (±25.7) brightly labeled HCs and 8.3 (±2.9) brightly labeled SCs per utricle (*Figure 5F–G'''*, *Figure 5—source data 1*). HCs were identified as myosin VIIa-positive cells with nuclei in the apical two-thirds of the epithelium (*Figure 5F–F'''*). SCs were identified as myosin VIIa-negative cells whose bodies extend across the entire macular depth, whose nuclei are smaller than HC nuclei, and are positioned near the basal lamina (*Figure 5G–G'''*). We also detected 8.0 (±2.6) brightly labeled cells per utricle that did not meet criteria for HCs or SCs. Often, these cells were myosin VIIa-negative with a nucleus located

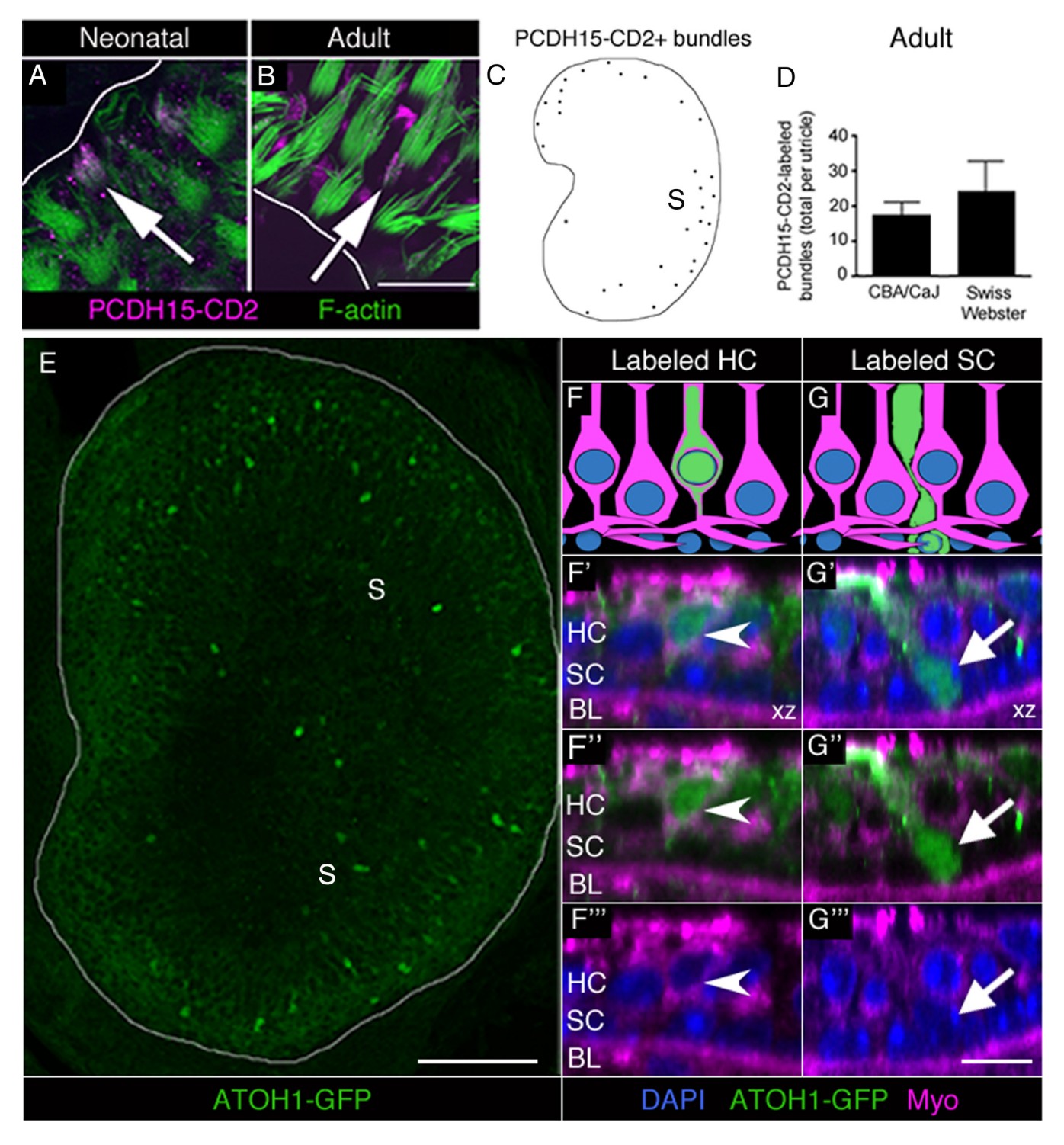

**Figure 5.** Immature HCs are found in adult mouse utricles under normal conditions. (**A,B**) Confocal xy optical sections of utricles from neonatal (<postnatal day 8) (**A**) and adult (>6 weeks) (**B**) Swiss Webster mice. Some HC stereocilia bundles at each age (arrows) were co-labeled with F-actin (green) and antibodies to PCDH15-CD2 (magenta). Some PCDH15-CD2 labeling occurred in other places on the epithelial surface, but our analysis was restricted to stereocilia only. White lines indicate the approximate location of edge of the utricular macula. (**C**) Schematized map of a utricle from a representative adult Swiss Webster mouse. HCs with PCDH15-CD2-positive stereocilia are depicted by black dots. S, striola. (**D**) Number of PCDH15-CD2-expressing HC stereocilia bundles per utricle in normal adult CBA/CaJ and Swiss Webster mice, where there was no statistically significant difference (determined by an unpaired two-tailed Student's *t*-test; p=0.1635, see *Figure 5—source data 1*). Data are expressed as mean +1 standard

*Figure 5 continued on next page*

*Figure 5 continued*

deviation. (**E**) Confocal projection image through the utricular macula of a 6-week-old *Atoh1GFP/GFP* mouse expressing ATOH1-GFP fusion protein (green). S, striola. (**F,G**) Schematics of cross-sectional views through a 6-week-old *Atoh1GFP/GFP* mouse utricle shown in **F′–F′″** and **G′–G′″**. (**F′–F″′**) Confocal optical section of a HC [arrowhead; myosin VIIa (Myo), magenta; DAPI, blue] that expressed ATOH1-GFP (green). (**G′–G″′**) Confocal optical section of a SC (arrow; DAPI, blue) that expressed ATOH1-GFP (green). HC, HC nuclear layer; SC, SC nuclear layer; BL, basal lamina. Scale bar in **B** is 12 μm and applies to **A**,**B**. Scale bar in **E** is 100 μm. Scale bar in **G′″** is 10 μm and applies to **F′–G′″**.

The following source data is available for figure 5:

**Source data 1.** Quantification of immature HC markers in the normal adult mouse utricle.

between the SC and HC nuclear layers and had thick apical necks. We postulate that these cells were SCs in the process of transitioning to a HC fate.

Collectively, these observations demonstrate that immature HCs exist in adult mouse utricles under normal conditions.

## SCs give rise to new type II HCs in adult mouse utricles under normal conditions

The presence of immature HCs in utricles from normal adult mice is consistent with the hypothesis that new HCs are added in adulthood. Based on previous work (*Lin et al., 2011*; *Burns et al., 2012*), the likely source of new HCs is neighboring SCs. To determine if utricular HCs derive from SCs in normal adult mice, we used *Plp1-CreERT2:ROSA26tdTomato* mice, in which tamoxifen induces tdTomato expression in SCs (*Figure 4B*, *Figure 4—source data 1*). Prior to fate-mapping, we determined that 91.7% (±6.1%; n = 3) of extrastriolar SCs were tdTomato-positive one week after tamoxifen injection (*Figure 4—source data 1*). We hypothesized that most new HCs would be added in this region, since we detected most HCs with immature markers (PCDH15-CD2 and ATOH1-GFP) there (*Figure 5C,E*). At 15 weeks post tamoxifen, the latest time analyzed in this fate-mapping study, 85.9% (±3.2%; n = 3) of extrastriolar SCs were labeled (*Figure 4—source data 1*). The concentration of *Plp1-CreERT2* activity in the extrastriola throughout the time-course of our experiment indicated that this mouse line could effectively record the transition of SCs into HCs.

To fate-map the transition of SCs into HCs during adulthood, we gave tamoxifen to 6-week-old *Plp1-CreERT2:ROSA26tdTomato* mice and examined utricles at 1, 4, 10, or 15 weeks later (*Figure 6A*). We assessed whether the number of tdTomato-positive HCs increased over time after induction of tdTomato labeling, as predicted if SCs convert into HCs. Utricles were labeled with antibodies to myosin VIIa and DAPI, and we collected optical slices throughout the entire macula using confocal microscopy. tdTomato-positive HCs (*Figure 6B–C′″*) were identified in all utricles and were scattered throughout the macula, but they appeared to be most common in the extrastriola (*Figure 6F*), which is consistent with findings for immature HC markers (*Figure 5C,E*).

We counted tdTomato-positive HCs in each utricle that were type I or type II at 1, 4, 10, or 15 weeks post tamoxifen using several criteria (see Materials and methods, *Video 4*, *Figure 1D*). Any cells that did not match criteria were scored as 'unknown.' Examples of tdTomato-positive type I and II HCs are shown in *Figure 6B–B″′* and *Figure 6C–C″′*, respectively. At one week post tamoxifen, we detected 4.0 (±4.5) tdTomato-positive type I and 8.5 (±4.8) tdTomato-positive type II HCs per utricle (*Figure 6D*, *Figure 6—source data 1*), which were not significantly different from age-matched no-tamoxifen controls (*Figure 6D*, *Figure 6—source data 1*). In tamoxifen-treated mice, the number of tdTomato-positive type II HCs per utricle increased over time from 8.5 (±4.8) cells at one week post tamoxifen to 36.6 (±7.7) cells at 15 weeks post tamoxifen (p<0.0001 for effect of treatment/time post tamoxifen; p<0.0001 for effect of HC type, determined by two-way ANOVA followed by Tukey's multiple comparisons test). However, there was no significant change in numbers of tdTomato-positive HCs that were type I or 'unknown' during this time (*Figure 6D*, *Figure 6—source data 1*). Linear regression analysis showed that the average number of tdTomato-positive type II HCs increased by ~2 cells per week (*Figure 6E*). In mice that did not receive tamoxifen, there was no significant change in tdTomato-positive HCs of any type over time (*Figure 6D*).

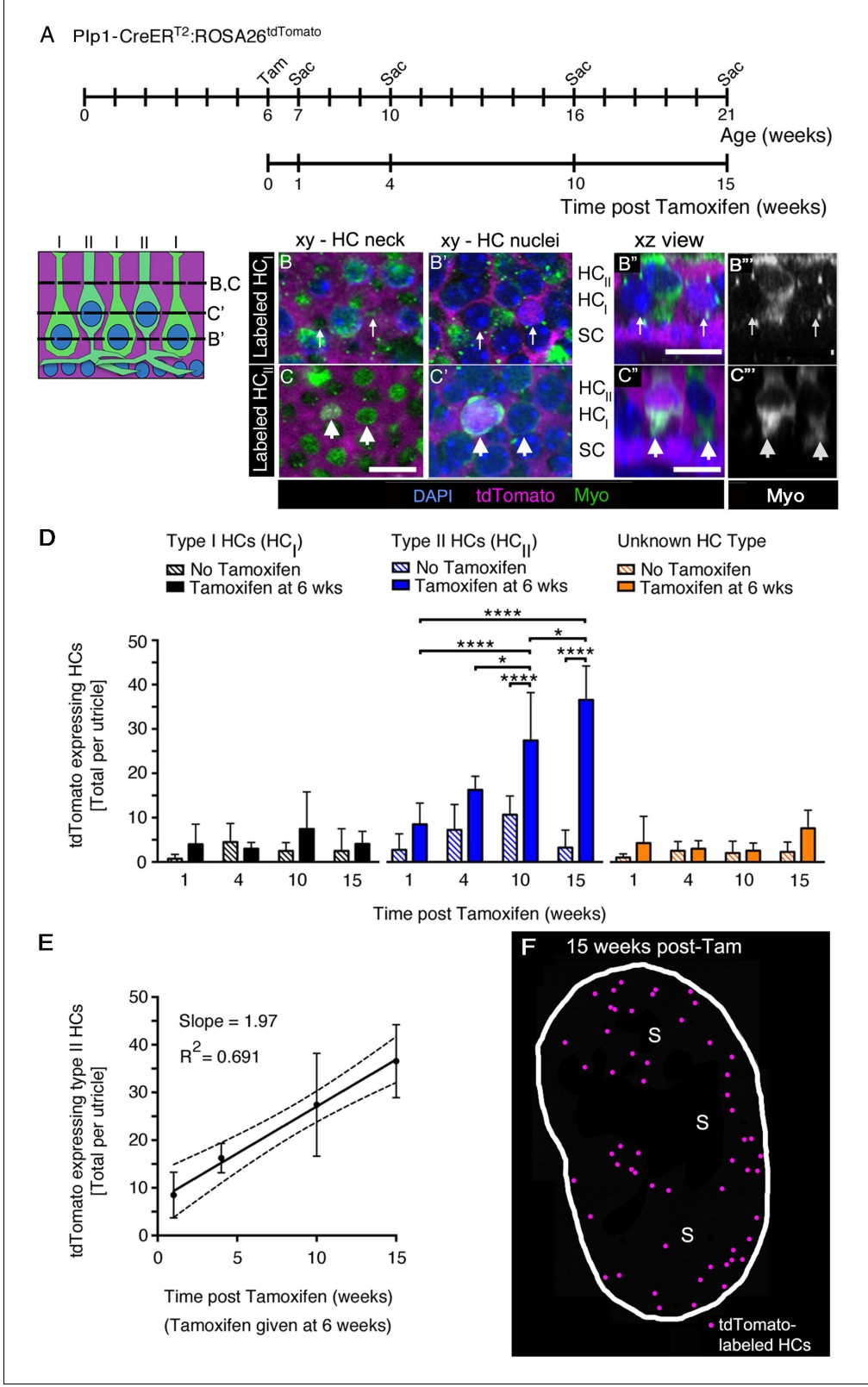

**Figure 6.** SCs produce new type II HCs in adult mouse utricles under normal conditions. (**A**) Experimental timeline. *Plp1-CreER^{T2}:ROSA26^{tdTomato}* mice were injected with tamoxifen (Tam) at 6 weeks of age to label SCs with tdTomato and were sacrificed (Sac) at 7, 10, 16, and 21 weeks of age (corresponding to 1, 4, 10, and 15 weeks post tamoxifen). Control animals (age-matched *Plp1-CreER^{T2}:ROSA26^{tdTomato}* mice that did not receive tamoxifen) were sacrificed at the same ages. (**B–C'''**) Examples of a tdTomato-labeled (magenta) type I HC (**B–B'''**) and type II HC (**C–C'''**) from a *Plp1-*

*Figure 6 continued on next page*

*Figure 6 continued*

*CreER^{T2}:ROSA26^{tdTomato}* mouse utricle at 10 weeks post tamoxifen, which were classified according to criteria defined in Materials and methods, *Figure 1D*, and *Video 4*. (B,B' and C,C') xy slices taken at the levels indicated in the schematic to the left. Myosin VIIa (Myo) is in green and DAPI is in blue. Thin arrows in B,B' point to two type I HCs, at the level of the neck (B) and the nucleus (B'). Only the type I HC on the right is tdTomato-positive, which is most evident in its nucleus (B'). Fat arrows in C,C' point to two type II HCs, at the level of the neck (C) and the nucleus (C'). Only the type II HC on the left is tdTomato-positive. (B",C") Xz view of the same cells shown in B–B' and C–C', providing perspective on their morphology and lamination. Labels indicate the approximate positions of the nuclei for each cell type (HC_{II}, type II HC; HC_{I}, type I HC; and SC, SC). (B"',C"'): Same images as B", C", but with Myo labeling only. Scale bar in C is 10 μm and applies to B–C'. Scale bars in B",C" are 10 μm and apply respectively to B"' and C"'. It is important to note that, in thin optical slices such as these, myosin VIIa labeling intensity varied across cells, independent of tdTomato labeling intensity. For example, in panels C" and C"', the type II HC on the left is brighter than the type II HC on the right. Further, in panels B',B", the myosin VIIa labeling for type I HC perinuclear cytoplasm was relatively weak, and labeling at the neck (B) was more robust. In cases such as this one, other morphological criteria—nuclear position, relative neck thickness, and presence/absence of a basolateral process—were essential for cell-typing. (D) Total number of tdTomato-expressing HCs per utricle, categorized by HC type [type I (black), type II (blue), and 'unknown' (orange)]. Patterned bars = control *Plp1-CreER^{T2}:ROSA26^{tdTomato}* mice that did not receive tamoxifen. Solid bars = *Plp1-CreER^{T2}:ROSA26^{tdTomato}* mice that received tamoxifen at 6 weeks of age. Data are expressed as mean ±1 standard deviation for n = 4–8 mice (see *Figure 6—source data 1*). *p<0.05; ****p<0.0001 as determined by a two-way ANOVA (p<0.0001 for treatment/age; p<0.0001 for HC type) followed by Tukey's multiple comparisons post-test. (E) Linear regression analysis of tdTomato-expressing type II HCs per utricle demonstrated an increase in labeled cells with time (data correspond to D, blue solid bars). The calculated slope was 1.97 type II HCs per week, and the R^2 value was 0.691. Data are expressed as mean ±1 standard deviation with dotted lines representing the 95% confidence interval. (F) Map of a representative *Plp1-CreER^{T2}:ROSA26^{tdTomato}* utricle injected with tamoxifen at 6 weeks of age and analyzed 15 weeks later. tdTomato-labeled HCs are depicted by magenta dots. S, striola.

The following source data is available for figure 6:

**Source data 1.** Quantification of tdTomato-labeled HCs in *Plp1-CreER^{T2}:ROSA26^{tdTomato}* utricles over time.

## Dividing SCs are rare in adult mouse utricles under normal conditions

To test if new HCs derive from division of SCs, we administered the thymidine analog 5'-bromo-deoxyurdine (BrdU) to normal adult mice. BrdU was delivered using two different routes of adminis-tration (intraperitoneal injection and drinking water) and different dosing regimens for up to two weeks of continuous administration. We examined utricles after BrdU treatment from 18 adult mice, including both C57Bl/6J and Swiss Webster strains (*Table 1*). We routinely detected strongly BrdU-labeled nuclei in within-animal positive control tissue (intestinal epithelium and utricular connective tissue), but we found only one example of a BrdU-labeled pair of cells in the utricular macula (not shown). Therefore, if SCs divide during HC turnover, they do so rarely. These observations are consistent with the slow rate of HC addition that we observed and with previous studies of cell division in the macula of normal mice (*Li and Forge, 1997*; *Kuntz and Oesterle, 1998*).

## *Atoh1-CreER^{TM}*-labeled type II HCs did not convert into type I HCs under normal conditions

Our observations indicate that, although type I and type II HCs are cleared from the macula under normal conditions, only type II HCs are replaced by *Plp1-CreER^{T2}*-expressing SCs. This finding raised the question of how type I HCs are maintained in adulthood. We hypothesized that type II HCs might convert into type I HCs. To test

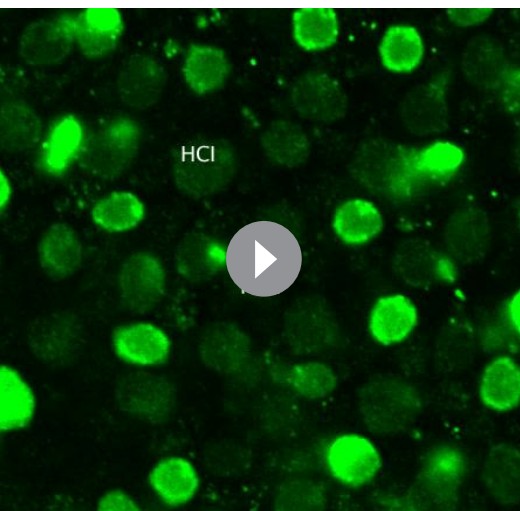

**Video 4.** Clearly stratified type I and type II HCs in the adult mouse utricle. This movie begins at the cuticular plate of the HCs, proceeds through the level of HC necks, type II HC nuclei (HCII), type I HC nuclei (HCI), and ends in the SC nuclear layer where type II HC basolateral processes are located. An example of a type II HC basolateral process is shown by the arrow. Myosin VIIa antibodies were used to label all HCs (green).

**Table 1.** Detection of utricular SC proliferation under normal conditions. We used two delivery methods and 3 different time periods of exposure to administer BrdU to mice to investigate cell proliferation in the utricular sensory epithelium (SE) in two strains of mice (C57Bl/6J and Swiss Webster) that were older than 6 weeks. BrdU was either injected intraperitoneally (IP) or administered per os (PO) in the drinking water. SD, one standard deviation.

| Number of BrdU+ cells per SE Mean (SD) | Delivery route | Delivery rate | Dose | Days of exposure | Number of mice/utricles (strain) |
|---|---|---|---|---|---|
| 0.25 (0.50) | IP | 1x/day | 50 mg/kg | 7 | 2/4 (C57Bl/6J) |
| 0 (0) | IP | 2x/day | 50 mg/kg | 3 | 3/3 (Swiss Webster) |
| 0 (0) | PO | Continuous | 2 mg/ml | 7 | 7/14 (C57Bl/6J) |
| 0 (0) | PO | Continuous | 2 mg/ml | 7 | 3/6 (Swiss Webster) |
| 0 (0) | PO | Continuous | 2 mg/ml | 14 | 3/6 (Swiss Webster) |

this hypothesis, we fate-mapped type II HCs under normal conditions in adult utricles using *Atoh1-CreER^TM^:ROSA26^tdTomato^* mice in which CreER expression is driven by an *Atoh1* enhancer (*Chow et al., 2006*). After tamoxifen administration to 6-week-old *Atoh1-CreER^TM^:ROSA26^tdTomato^* mice, we used myosin VIIa immunolabeling and cell classification criteria (*Figure 1D*, Materials and methods, *Video 5*) to determine that no SCs were tdTomato-positive at one week post tamoxifen. Using the same criteria, we determined that 92.7% (±2.8%) of tdTomato-labeled cells were type II HCs and 5.9% (±3.0%) were type I HCs (*Figure 7C–F"*, *Figure 7—source data 1*). Assuming a total utricular HC population of 3800 (*Golub et al., 2012*) and a type I:type II ratio of 1.17:1 (*Pujol et al., 2014*), we estimate that ~2.5% of type I HCs and ~39.3% of type II HCs were labeled in *Atoh1-CreER^TM^:ROSA26^tdTomato^* mouse utricles. Age-matched control mice that did not receive tamoxifen had fewer than 15 labeled HCs per utricle, indicating a low level of Cre activity in the absence of tamoxifen (*Figure 7B*, *Figure 7—source data 1*).

Next, we tested if the number of tdTomato-labeled type I HCs increased over time, as one would predict if significant numbers of labeled type II HCs were to convert into type I HCs. We administered tamoxifen to *Atoh1-CreER^TM^:ROSA26^tdTomato^* mice at 6 weeks of age and euthanized them at 10, 15, or 32 weeks post tamoxifen (*Figure 7A*). Control mice that did not receive tamoxifen were euthanized at 16, 21, and 38 weeks of age (*Figure 7A*). Using analysis of covariance (ANCOVA) to control for variability in the total number of tdTomato-labeled cells across timepoints, we found no significant change in tdTomato-labeled type I HCs between 1 and 32 weeks post tamoxifen (p=0.103; *Figure 7G*, *Figure 7—source data 1*). These observations demonstrate that few, if any, type II HCs labeled in *Atoh1-CreER^TM^:ROSA26^tdTomato^* mice transdifferentiated into type I HCs during the 8-month period we examined.

## Diphtheria toxin-induced HC ablation increased SC-to-type II HC transition in adult mouse utricles

Our results demonstrate that SCs transdifferentiate into type II HCs at a low rate under normal conditions. To determine if SC transdifferentiation is increased after destruction of HCs, we bred *Plp1-CreER^T2^:ROSA26^tdTomato^* mice with *Pou4f3^DTR^* mice, in which the human diphtheria toxin receptor (DTR) is knocked into the endogenous coding region for *Pou4f3*, a HC-specific transcription factor (*Golub et al., 2012*). Injection of diphtheria toxin (DT) to mice with a single *Pou4f3^DTR^* allele kills all but 6% of HCs in the utricle, and over the next two months, HC numbers are restored to 17% of normal levels (*Golub et al., 2012*). This study found that only type II HCs are regenerated. Since type II HCs constitute half of the normal HC population, we assume that ~34% of the type II HC population is regenerated.

We injected *Plp1-CreER^T2^:ROSA26^tdTomato^:Pou4f3^DTR^* mice with tamoxifen at 6 weeks of age, then with DT one week later (*Figure 8A*). Utricles were collected and labeled with antibodies against myosin VIIa at 3 weeks post DT (equivalent to 4 weeks post tamoxifen). We expected, based on our

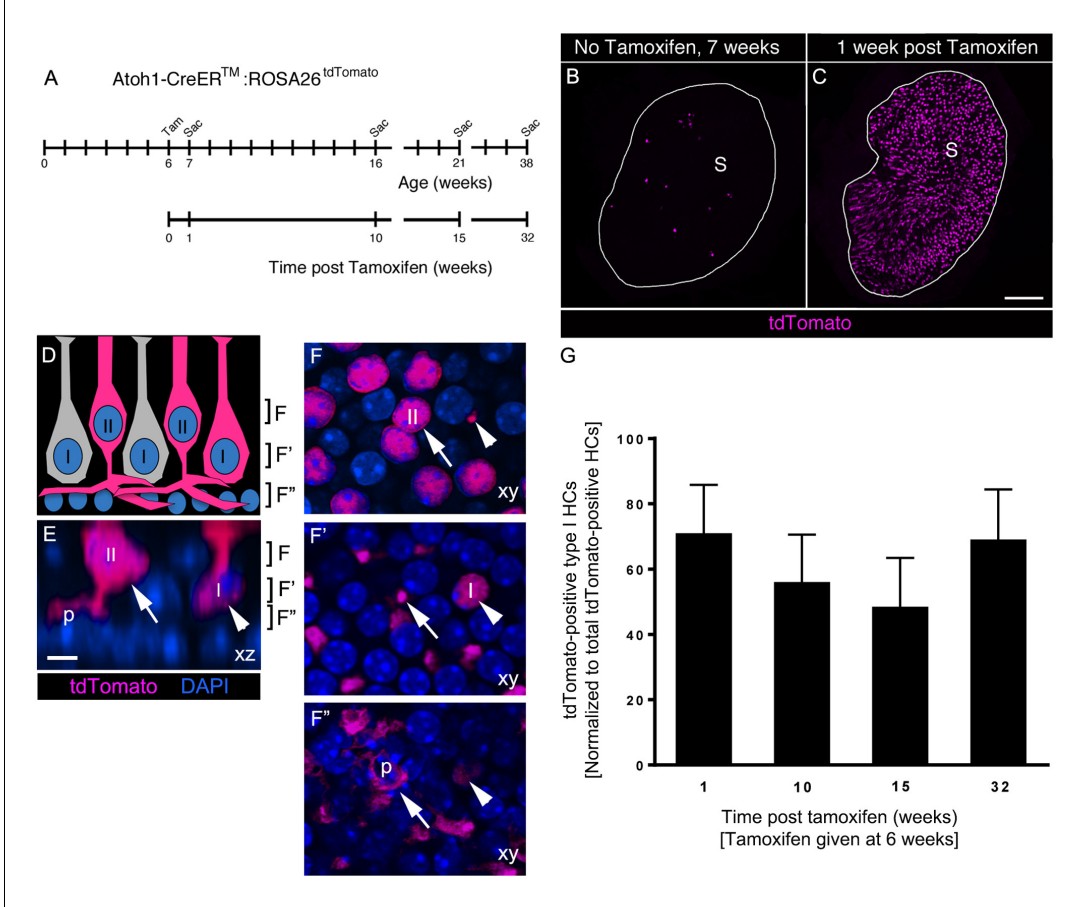

**Figure 7.** *Atoh1-CreER*[TM]-labeled type II HCs do not convert into type I HCs over 8 months in adult mouse utricles under normal conditions. (**A**) Experimental timeline. *Atoh1-CreER*[TM]: *ROSA26*[tdTomato] mice were injected with tamoxifen (Tam) at 6 weeks of age and were sacrificed (Sac) at 7, 16, 21, and 38 weeks of age (corresponding to 1, 10, 15, and 32 weeks post tamoxifen). (**B,C**) Confocal projection images of whole utricles from 7-week-old *Atoh1-CreER*[TM]:*ROSA26*[tdTomato] mice showing the numbers and distribution of tdTomato-positive SCs (magenta) in the utricular sensory epithelium of mice that received no tamoxifen (**B**) or mice that received tamoxifen (**C**) S, striola. (**D–F"**) Examples of tdTomato-labeled type I and type II HCs. (**D**) A schematic of a cross-section through the utricular macula, with brackets indicating the optical sections generated for **F–F"**. I, type I HC; II, type II HC. (**E**) Xz view (similar to **D**) of images used to generate **F–F"**, taken from the extrastriolar region of an *Atoh1-CreER*[TM]:*ROSA26*[tdTomato] mouse at one week post tamoxifen. Brackets indicate the location of optical sections shown in **F–F"**. Arrow points to a tdTomato-positive type II HC (II) with a basolateral process (p). Arrowhead points to a tdTomato-positive type I HC (I) with a thin neck and more basally located nucleus than the type II HC. tdTomato is shown in magenta; DAPI is shown in blue. (**F–F"**) Progressively deeper optical xy sections through the utricular macula. Arrows and arrowheads point to same HCs as shown in **E**. Note that the type II HC has an apically located nucleus (arrow, [**E,F**]) and a basolateral process (arrows, **E,F',F"**; p, **E,F"**). Note that the type I HC (arrowhead, [**E**]) has a thin neck (arrowhead, [**F**]), a basally located nucleus (arrowhead, [**F'**]) and no basolateral process (arrowhead, [**F"**]). Scale bar in **C** is 100 µm and applies to **B,C**. Scale bar in **E** is 6 µm and applies to **E–F"**. (**G**) Number of tdTomato-positive type I HCs at 1, 10, 15, and 32 weeks post tamoxifen normalized to the total number tdTomato-positive cells at each timepoint (see *Figure 7–source data 1* for raw data). No significant differences in tdTomato-positive type I HCs were observed across time (determined by ANCOVA, p=0.103; n = 4). Data are expressed as mean ±1 standard deviation.

The following source data is available for figure 7:

**Source data 1.** Quantification of tdTomato-labeled HCs in *Atoh1-CreER*[TM]:*ROSA26*[tdTomato] utricles.

prior study (*Golub et al., 2012*), that HC regeneration would be well underway at this point and we would detect tdTomato-labeled HCs if SCs generated new HCs. Age-matched *Plp1-CreER*[T2]: *ROSA26*[tdTomato] mice (lacking the *Pou4f3*[DTR] allele) were treated identically and served as negative controls, since DT causes no vestibular HC loss when administered to mice lacking the *Pou4f3*[DTR] allele (*Golub et al., 2012*).

Utricles from control mice showed normal-appearing HC numbers (*Figure 8B,D–D'*). By contrast, utricles from *Plp1-CreER^T2:ROSA26^tdTomato:Pou4f3^DTR* mice had a large reduction in HC numbers after DT treatment (*Figure 8C,E–E'*). We counted 424.8 (±119.0; 95% confidence interval: 308.1–541.4; n = 4) HCs per utricle (~12% of controls), the vast majority of which were type II, as expected. When we scored HCs for tdTomato labeling, we found that significantly more tdTomato-labeled type II HCs were present in damaged versus control utricles (*Figure 8F*; p=0.0346; unpaired, two-tailed Student's *t*-test). During the 4-week period after tamoxifen injection, SCs generated 6 times more tdTomato-labeled type II HCs in damaged utricles (101.3 ± 49.7) compared to controls (16.3 ± 4.5) (*Figure 8F*, *Figure 8—source data 1*). Assuming a HC population of 3800 (*Golub et al., 2012*), 0.4% (±0.1%) of HCs were tdTomato-labeled in control utricles. In contrast, 23.9% (±8.2%) of HCs were tdTomato-labeled in damaged utricles (*Figure 8—source data 1*). These results demonstrate that SCs in adult mouse utricles mount a truly regenerative response to damage, significantly increasing transdifferentiation into type II HCs relative to baseline levels.

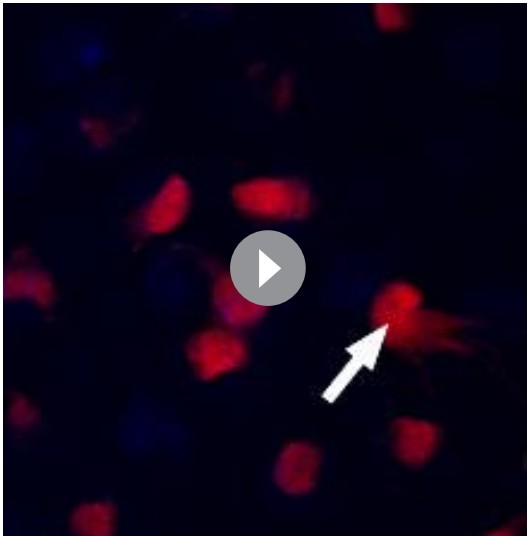

**Video 5.** Type I and type II HCs labeled with *Atoh1-CreER^TM:ROSA26^tdTomato*. This movie was constructed from optical sections collected from the area shown in *Figure 7E–F''*. The movie begins at the cuticular plate, passes through HC necks, type II HC nuclei, type I HC nuclei, SC nuclei, and ends at the basal lamina. About 10 type II HCs and one type I HC (arrow) are labeled with tdTomato (red). All nuclei are labeled with DAPI (blue). At the level of the type II HC nuclei, the type I HC neck (arrow) can be seen. At the level of the SC nuclei, tdTomato-labeled type II basolateral processes are observed.

## F-actin-rich phagosomes do not increase proportionately with dying HCs, or associate spatially with dying cells, after DT-induced damage

Our findings raised the question: do mechanisms of HC clearance following DT-induced HC damage resemble those during normal conditions? To address this, we labeled utricles from *Pou4f3^DTR* mice at 4, 7, 14, 40, 90, or 120 days post DT, as well as from littermates that lacked the *Pou4f3^DTR* allele (labeled as 0 day post DT) with antibodies to myosin VIIa, TUNEL, and DAPI (*Figure 9A–D*). Degenerating HCs were abundant at 4 and 7 days post DT (*Figure 9B–C*), as described previously (*Golub et al., 2012*). Degenerating HCs, seen in their normal positions, had myosin VIIa in their nuclei, which was not observed in utricles under normal conditions (*Figure 9A–B''*). Further, degenerating HCs had nuclei with apoptotic features: some nuclei had abnormal shapes and/or condensed chromatin (*Figure 9B–B''*), and some nuclei were TUNEL-positive (*Figure 9C*). By contrast, HCs that were being cleared under normal conditions were ectopic (located near the basal lamina), and most had healthy-appearing nuclei (*Figure 1E–E'''*).

Somewhat surprisingly, few TUNEL-positive nuclei seen after DT treatment were associated with F-actin-rich phagosomes (*Figure 9C*). The number of phagosomes per utricle increased significantly over time (p=0.0001; one-way ANOVA; n = 3–7), with 4 and 7 days post DT showing a ~60% increase relative to controls (p<0.05; Dunnett's multiple comparisons test). Phagosome numbers returned to control levels at later times post DT (*Figure 9D*, *Figure 9—source data 1*). This increase was much lower than expected if F-actin-rich phagosomes play a major role in HC clearance after DT-induced HC death, since we previously showed a decrease of >2000 HCs during the first 7 days post DT in *Pou4f3^DTR* mice (*Golub et al., 2012*).

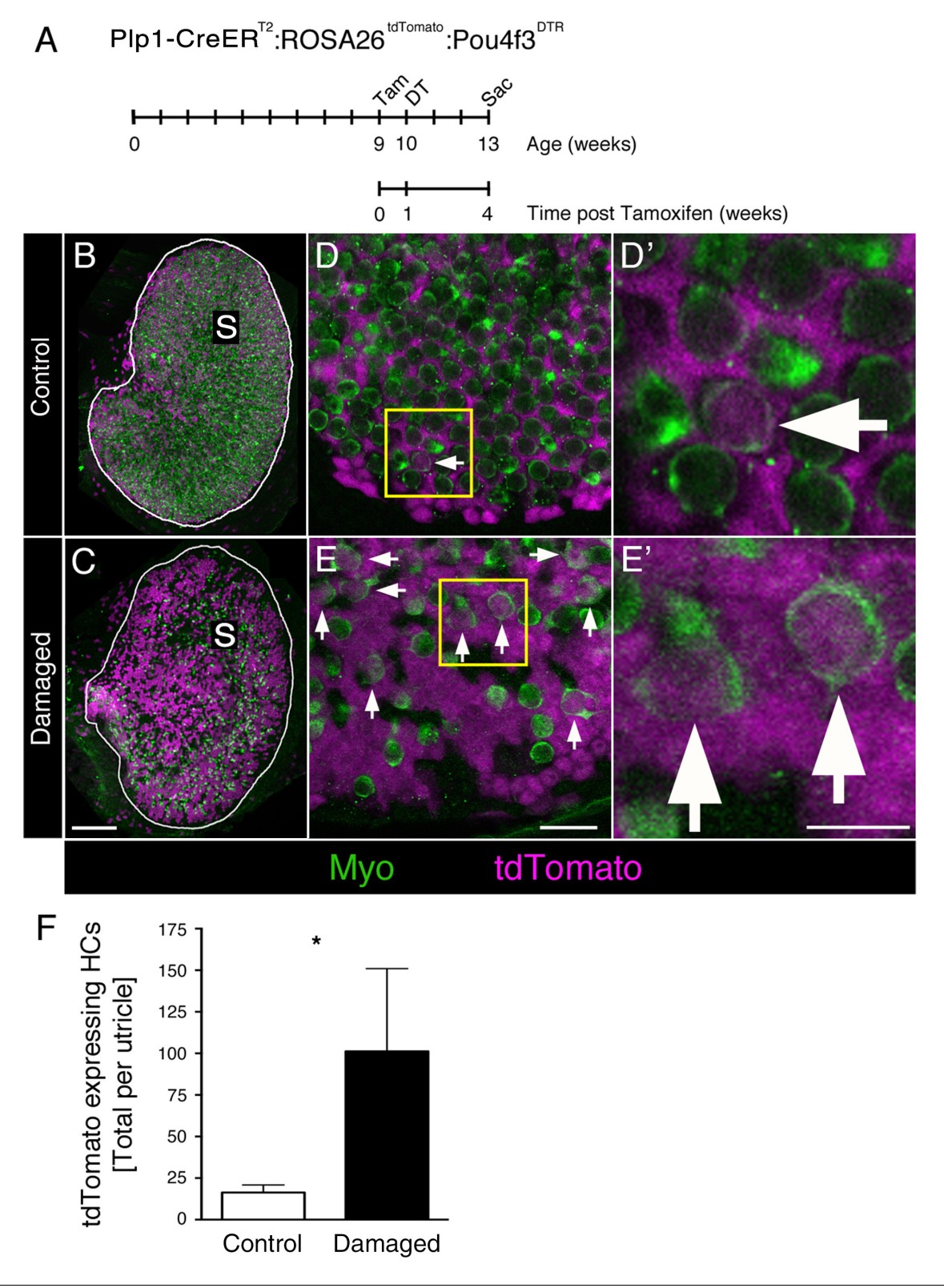

**Figure 8.** DT-mediated HC damage increases SC-to-HC transition in adult mouse utricles. (**A**) Experimental timeline. *Plp1-CreER^T2:ROSA26^tdTomato:* *Pou4f3^DTR* (damaged) mice and *Plp1-CreER^T2:ROSA26^tdTomato* (control) mice were injected with tamoxifen (Tam) at 9 weeks of age to label SCs with tdTomato, injected with diphtheria toxin (DT) at 10 weeks of age to kill HCs, and sacrificed (Sac) at 13 weeks of age (corresponding to 4 weeks post tamoxifen and 3 weeks post DT). (**B**) Utricles from control mice (*Plp1-CreER^T2:ROSA26^tdTomato*) that received DT injection but lacked the *Pou4f3^DTR*

*Figure 8 continued on next page*

*Figure 8 continued*

allele exhibited normal-appearing HC densities [myosin VIIa (Myo), green]. S, striola. (**C**) Utricles from damaged *Plp1-CreER^{T2}:ROSA26^{tdTomato}:Pou4f3^{DTR}* mice sacrificed 3 weeks post DT had fewer HCs (Myo, green). SCs were labeled with tdTomato (magenta) in both control and damaged adult mouse utricles (**B,C**). S, striola. (**D,E**) Confocal optical images of the extrastriolar region from similar utricles as shown in **B** and **C**, acquired at higher magnification. (**D′,E′**) Higher magnifications of the boxed areas in **D,E**. tdTomato-expressing HCs (arrows) were detected in control (**D,D′**) and damaged (**E,E′**) utricles. Scale bar in **C** is 100 µm and applies to **B,C**. Scale bar in **E** is 20 µm and applies to **D,E**. Scale bar in **E′** is 10 µm and applies to **D′,E′**.( **F**) Total number of tdTomato-expressing HCs at 13 weeks of age (equivalent to 3 weeks post DT and 4 weeks post tamoxifen) in control (white bar, n = 3) and damaged (black bar, n = 4) utricles (see *Figure 8—source data 1*). Damaged utricles had significantly more tdTomato-labeled HCs compared to control utricles determined by an unpaired, two-tailed Student's *t*-test (p=0.0346). Data are expressed as mean ±1 standard deviation.

The following source data is available for figure 8:

**Source data 1.** Quantification of tdTomato-labeled HCs after HC damage in *Plp1-CreER^{T2}:ROSA26^{tdTomato}:Pou4f3^{DTR}* and control utricles.

## Discussion

The vestibular epithelia of the inner ear are composed of sensory HCs and non-sensory SCs that are derived from the otic placode. Our data show that, in utricles of adult mice, SCs remove type I and type II HCs under normal physiological conditions by phagocytosis and replace type II HCs by trans-differentiation. This study provides definitive evidence that vestibular epithelia are a site of ongoing HC turnover in adult mammals. In this sense, vestibular epithelia resemble other placodally derived neurosensory epithelia, such as taste buds (*Okubo et al., 2009*) and olfactory epithelia (*Leung et al., 2007*), which cull and replace sensory receptors throughout life. However, vestibular epithelia contrast sharply with the organ of Corti, the sensory organ of hearing, which is also derived from the otic placode, but in which no new HCs are formed under normal circumstances or after damage in adult mammals (*Bohne et al., 1976*; *Hawkins et al., 1976*; *Forge et al., 1998*; *Oesterle et al., 2008*; *Cox et al., 2014*).

### Phagocytosis by SCs mediates HC clearance during normal cell turnover in adult mouse utricles

SCs, which are epithelial cells that resemble glia in several respects, serve many functions in HC epithelia, including maintaining structural integrity, clearing ions and neurotransmitters from extracellular space, and releasing neurotrophins to maintain synapses between HCs and neurons (reviewed in *Monzack and Cunningham, 2013*; *Wan et al., 2013*). SCs also phagocytose HCs that are injured by ototoxins or intense noise (*Abrashkin et al., 2006*; *Bird et al., 2010*; *Anttonen et al., 2014*). Our results demonstrate an additional, previously unknown role of SCs as phagocytes of type I and II vestibular HCs under normal conditions. While data from both *Plp1-CreER^{T2}:ROSA26^{tdTomato}* and *Lfng-eGFP* mice support the hypothesis that SCs act as phagocytes to remove HCs, we cannot rule out that other cell types in the organ also play this role. For instance, Schwann cells were also labeled by tdTomato in utricles of *Plp1-CreER^{T2}:ROSA26^{tdTomato}* mice, and they have phagocytic capabilities in the peripheral nervous system (*Band et al., 1986*; *Reichert et al., 1994*). We attempted to assess Schwann cell contribution to phagosomes, but we were unable to identify antibodies or CreER mouse lines that labeled Schwann cells without also labeling SCs. *Plp1-CreER^{T2}:ROSA26^{tdTomato}* mice also labeled cells in the transitional epithelium. However, these cells are unlikely sources of phagosomes, since the majority of phagosomes we observed were located in the central region of the macula, away from the transitional epithelium.

Our study did not determine the reason why HC clearance is concentrated centrally. This finding is somewhat counterintuitive, since HC addition appears to be most abundant in the periphery. However, it is possible that HC clearance and HC addition are not spatially coupled. During development, addition of utricular HCs in rodents occurs in a central-to-peripheral gradient (*Sans and Chat, 1982*; *Burns et al., 2012*). Since we found HC clearance to be focused centrally, we hypothesize that the oldest HCs are being removed. During adulthood, peripherally added HCs might migrate toward the center over time, where they are eventually cleared. These hypotheses remain to be tested.

Since our study involved analysis of static images, we were unable to determine the temporal sequence of events, how quickly HCs are removed, how many HCs are cleared by SCs per week, the

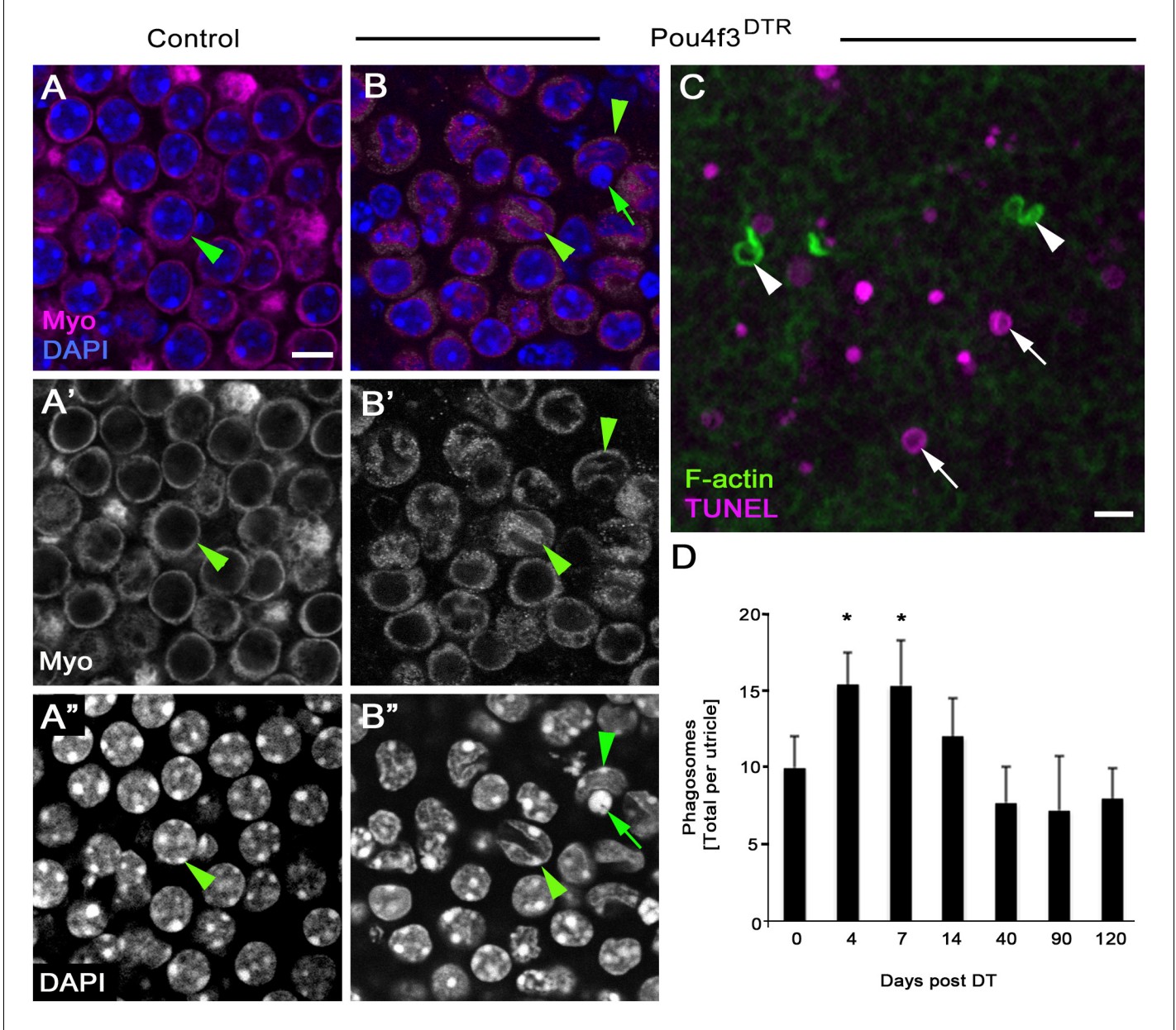

**Figure 9.** DT-mediated HC damage induces apoptosis and a small increase in phagosome numbers in adult mouse utricles. (**A–A"**) The same field of a control (*Pou4f3*[DTR]-negative) utricle, focused on the type II HC layer. (**B–B"**) The same field of a *Pou4f3*[DTR] utricle at 4 days post DT, focused on the HC layer. Green arrowheads point to myosin VIIa-labeled HCs (Myo, magenta in **A,B**; white in **A',B'**), while green arrows point to a nucleus (DAPI, blue in **B**; white in **B''**) with condensed chromatin. (**C**) Projection image of a *Pou4f3*[DTR] utricle at 7 days post DT. Arrowheads point to F-actin-rich (green) phagosomes, while arrows point to TUNEL-labeled (magenta) DNA. Scale bar in **A** is 6 μm and applies to **A–B"**. Scale bar in **C** is 5 μm. (**D**) Total number of phagosomes per *Pou4f3*[DTR] utricle at different times post DT. See *Figure 9–source data 1* for raw data. There were significantly more phagosomes at 4 and 7 days post DT compared to control littermates lacking the *Pou4f3*[DTR] allele (0 day post DT) determined by a one-way ANOVA ($p < 0.0001$) followed by a Dunnett's multiple comparisons test (*$p < 0.05$: n = 3–7). Data are expressed as mean +1 standard deviation.

The following source data is available for figure 9:

**Source data 1.** Quantification of phagosomes in *Pou4f3*[DTR] mice after HC damage.

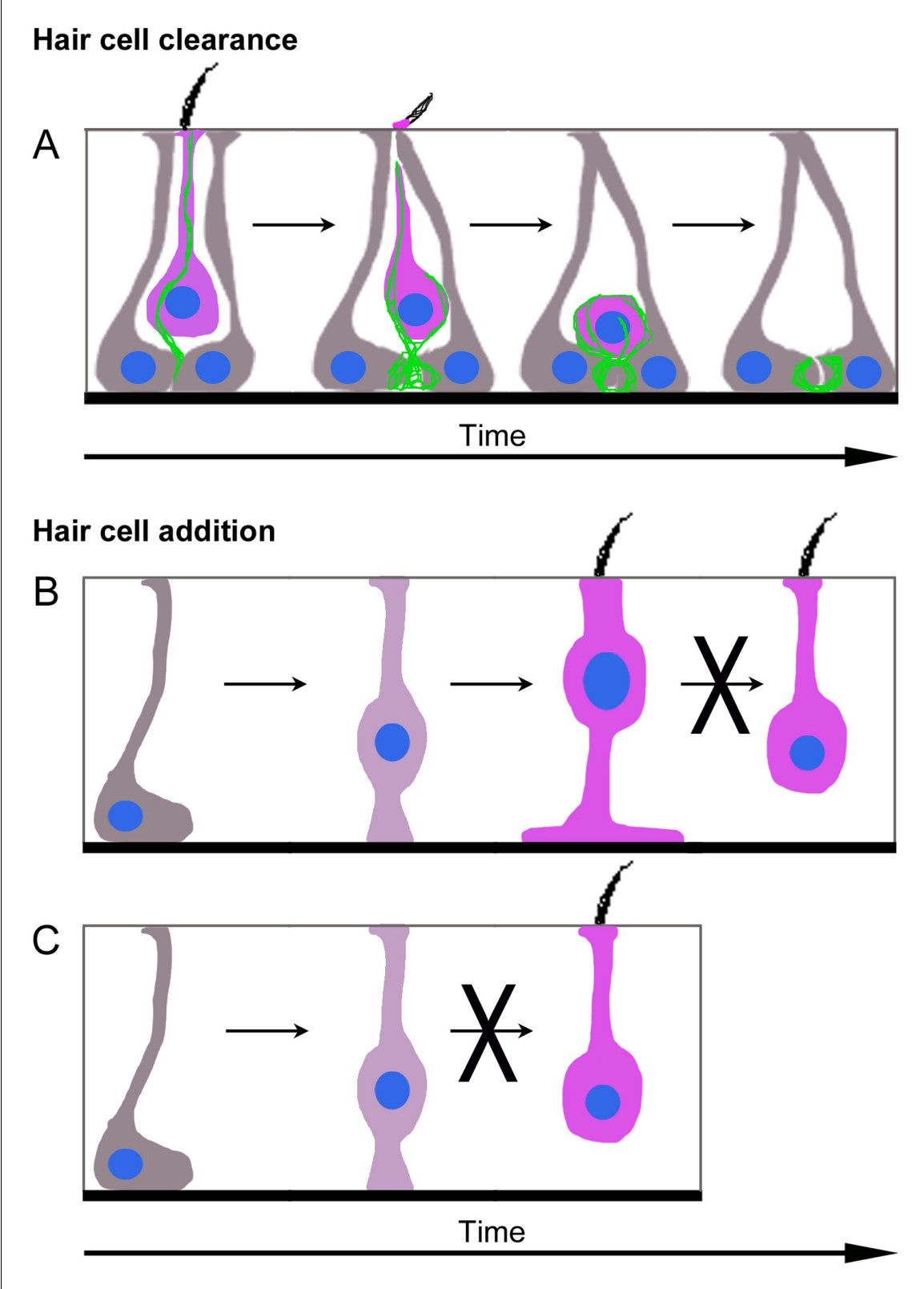

**Figure 10.** Model of HC turnover in adult mouse utricles under normal conditions. (**A**) Model of HC clearance. An actin spike (green) forms to connect SCs (grey) and a HC targeted for clearance (magenta). The apical portion of SCs converges and cleaves off the top of the HC, including the stereocilia. The actin spike aids the translocation of the HC to the SC nuclear layer, where SCs form an actin-rich phagosome and engulf the HC. Once the HC is removed from the sensory epithelium, an empty ring- or basket-like actin structure may remain until it is resorbed by SCs. (**B,C**) Model of type II HC

*Figure 10 continued on next page*

Figure 10 continued

addition. (B) *Plp1-CreER^{T2}*-expressing SCs (grey) give rise to type II HCs (magenta) by first translocating their nuclei into the HC layer and becoming an intermediate cell type (dark magenta). Type II HCs that express *Atoh1-CreER^{TM}* do not transdifferentiate into type I HCs. (C) *Plp1-CreER^{T2}*-expressing SCs (grey) do not directly generate type I HCs (magenta) in adulthood.

dynamics of phagosome formation, or the lifespan of phagosomes. However, based on our observations, we generated a model (*Figure 10A*) that should be tested using live-cell imaging and serial immuno-electron microscopy. We propose that the first sign a HC has been 'selected' for clearance is the appearance of a large actin spike in its cytoplasm. Actin-rich cables or 'cytocauds' are also observed in HC cytoplasm of adult utricles in mice (*Sobin et al., 1982*), guinea pigs (*Flock et al., 1979*; *Kanzaki et al., 2002*), and humans (*Taylor et al., 2015*). We suspect this actin spike is either derived from the HC itself or from a SC that has been signaled to 'attack' the HC. We hypothesize that this spike extends and becomes anchored basally, at which point the HC body is translocated to the basal compartment. Once translocated, adjacent SCs create basket-like phagosomes and consume the HC. However, it is also possible that the actin spike does not play a role in HC clearance. Because we did not detect stereocilia in translocated HCs, we presume that the apical portion of each HC is cleaved by SCs and ejected apically before translocation, similar to what occurs after aminoglycoside toxicity (*Meiteles and Raphael, 1994*; *Li et al., 1995*; *Bird et al., 2010*; *Monzack et al., 2015*). However, it is also possible that stereocilia proteins are degraded in HCs prior to, or during, translocation.

SC-derived phagosomes that clear HCs during turnover share some features with those that clear HCs after drug damage, including their high actin composition and shape. However in normal utricles, we found that HCs are pierced by a single actin spike and translocated to a basal position, which has not been reported after damage. Although the number of phagosomes increased after DT-mediated HC death, the increase was not proportionate to HC loss, and many apoptotic HCs were not localized near phagosomes. These observations suggest that F-actin-rich phagosomes derived from SCs play a minor role in removing HCs in mouse utricles after DT treatment, which is consistent with previous observations (*Kaur et al., 2015*). By contrast, F-actin-rich phagosomes derived from SCs actively clear utricular HCs following aminoglycoside treatments in vitro (*Bird et al., 2010*; *Monzack et al., 2015*). Further, we found that naturally occurring HC death does not trigger infiltration of immune-derived macrophages, which is robust after DT treatment (*Kaur et al., 2015*). Additional studies are needed to understand why different forms of HC degeneration trigger distinct forms of cell clearance.

It is not clear why certain HCs are removed from the epithelium under normal conditions. Most HCs associated with phagosomes appear normal aside from their unusual basal position. They have a typical nucleus and protein expression and show no nuclear condensation that occurs during apoptosis, nor cell swelling characteristic of necrosis (reviewed in *Nikoletopoulou et al., 2013*). Nonetheless, HCs may be phagocytosed because they become damaged due to normal 'wear and tear' under physiological conditions. Thus, it is important to define the triggers for HC clearance. These could include degeneration of HCs or changes in innervation, cell membrane integrity, or other HC features. Further, SCs might act as primary phagocytes (*Brown and Neher, 2012*; *Monzack and Cunningham, 2013*).

## SCs transdifferentiate into type II HCs during normal cell turnover in adult mouse utricles

We demonstrate that SCs play a second critical role in maintaining vestibular epithelia under normal conditions: they form new HCs, presumably replacing those lost during clearance. Consistent with this, small numbers of immature HCs were detected in normal utricles of adult mice using two markers: PCDH15-CD2 and ATOH1-GFP. Recently, evidence for immature stereocilia bundles was found in utricles of aged humans (>60 years old) (*Taylor et al., 2015*), suggesting HC turnover may also occur in adult primates. Two prior studies did not find evidence for ATOH1-immunolabeled cells (*Wang et al., 2010*) or cells with *Atoh1* enhancer activity (*Lin et al., 2011*) in normal utricles from adult mice. These contrasting results may be due to different sensitivities of the methods used to assess *Atoh1* expression.

Fate-mapping of SCs in *Plp1-CreER^{T2}:ROSA26^{tdTomato}* mice provided additional evidence for new HC addition. We assume SCs are the sole source of new HCs in our experiments based on a large body of literature showing that SCs are progenitors to vestibular HCs in non-mammalian vertebrates during both regeneration and turnover (reviewed in *Corwin and Oberholtzer, 1997*; *Warchol, 2011*). However, transitional epithelial cells and Schwann cells were also labeled in utricles of *Plp1-CreER^{T2}:ROSA26^{tdTomato}* mice. Therefore, additional CreER lines are needed to fate-map these cell populations and determine if they are unexpected sources of HCs during turnover.

Our data showed that SCs generate new type II HCs, but not type I HCs, under normal conditions (*Figure 10B,C*). These findings were surprising, since we detected clearance of both HC types. We hypothesized that type I HCs might be replaced by type II HCs since type II-to-I HC conversion has been suggested to occur after ototoxin-induced HC loss in avian utricles (*Weisleder and Rubel, 1993*; *Zakir and Dickman, 2006*). Additional support for this hypothesis was provided by *Kirkegaard and Jørgensen (2001)*, who detected individual HCs that appeared to be a hybrid with properties of both type I and type II HCs in vestibular organs of adult bats. However, when we fate-mapped type II HCs in mice under normal conditions, there was no evidence of type II-to-I conversion over an 8-month period of early adulthood (*Figure 10B*). However, the rate of conversion could take longer than 8 months. Further, type I HCs may derive from type II HCs that were not fate-mapped in *Atoh1-CreER^{TM}:ROSA26^{tdTomato}* mice or from SCs that were not fate-mapped in *Plp1-CreER^{T2}:ROSA26^{tdTomato}* mice. It should be noted, however, that only type II HCs are replaced after drug-induced damage in adult guinea pigs (*Forge et al., 1998*) and mice (*Kawamoto et al., 2009*; *Golub et al., 2012*), supporting the interpretation that type I HC replacement is attenuated or perhaps impossible in adult mice.

Regenerated HCs share many features with neighboring mature type II HCs, including myosin VIIa immunoreactivity, properly positioned nuclei, proper relative neck thickness, well defined stereocilia bundles, and basolateral processes. However, since we could not birth-date HCs in this study, we were not able to determine the degree to which new type II HCs are mature. Further, we did not perform any analyses that would inform on the functional status of the new HCs. Therefore, additional work is required to assess if HCs replaced in adult animals under normal conditions or after damage are functionally mature.

We were unable to distinguish whether HCs added during normal conditions derive from the progeny of SC division or via direct transdifferentiation, during which SCs phenotypically convert into HCs without dividing (reviewed in *Stone and Cotanche, 2007*). Using multiple methods for BrdU administration, we only detected BrdU labeling in rare cells within the utricular macula, and their identity was not established. This finding is consistent with other studies that assessed SC division in normal utricles *in vivo* (*Li and Forge, 1997*; *Kuntz and Oesterle, 1998*). In contrast, dividing SCs are readily detected during HC turnover in mature birds (*Roberson et al., 1992*; *Kil et al., 1997*; *Stone et al., 1999*). We cannot rule out, however, that SCs divide at a very low rate and our methods were insufficient to record them. Indeed, mammalian utricles that were cultured and treated with aminoglycoside antibiotics generated new HC-like cells and SCs by mitotic division (*Warchol et al., 1993*). One possible interpretation for the discrepancy is that culture conditions promote utricular SC division.

Prior fate-mapping studies showed that SCs form new vestibular HCs in juvenile and adult mice after HC damage (*Lin et al., 2011*; *Slowik and Bermingham-McDonogh, 2013*; *Wang et al., 2015*). In the present study, we found that, upon HC destruction, SC transdifferentiation into type II HCs increases by 6-fold over type II HC addition under normal conditions. This demonstrates that adult mammals are able to upregulate vestibular HC addition, relative to normal levels, upon HC injury.

## Significance of vestibular HC turnover in adult mice

We can use several approaches to estimate the rate of HC turnover. SC fate-mapping in mice with a mixed genetic background indicated that ~2 HCs are added to each utricle per week. This rate is likely a slight underestimate, since we only fate-mapped ~90% of extrastriolar and ~70% of striolar SCs in the utricle. Further, there could also be other sources of added HCs that were not fate-mapped, such as cells in the transitional epithelium or even in the stroma. We assessed the rate of HC addition by examining two markers for immature HCs (PCDH15-CD2 and ATOH1-GFP) in mice with a range of backgrounds. These markers labeled ~20 and ~82 HCs per utricle, respectively, which suggests a higher rate of HC addition than fate-mapping. However, we do not know how long

each marker is expressed in differentiating cells, so we cannot use these analyses to define the rate of HC addition.

We can also assess the rate of turnover by estimating how many HCs die in a given period. Swiss Webster mice had ~50 phagosomes per utricle, of which ~12 were associated with HCs, while *Plp1-CreER^T2:ROSA26^tdTomato* mice on a mixed background had ~28 phagosomes per utricle, of which ~5 were targeting HCs. Only 10–15 phagosomes were detected in CBA/CaJ or C57Bl/6J mice. It is unclear why Swiss Webster mice have significantly more phagosomes than the other tested mouse strains. They may have a higher rate of HC turnover, or their phagosomes may remain in the utricle for a longer period of time than in other strains. Although we cannot deduce the lifespan of a phagosome or the duration of HC clearance in the current study, *Monzack et al. (2015)* showed in aminoglycoside-treated mouse utricles that a phagosome engulfed a HC, cleared it, and began retracting within 8 hr. If this time frame is similar in normal utricles, and we use the conservative estimate from *Plp1-CreER^T2:ROSA26^tdTomato* (mixed background) mice that 5 HCs at any given time are being targeted by phagosomes generated within an 8 hr period, then we can estimate that one HC is targeted for removal every 1.6 hr, which is equivalent to 15 HCs per day or 105 HCs per week. Since numbers of HCs are maintained in adult mice, this could suggest that as many as 105 HCs are added per week, which is considerably more than indicated by SC fate-mapping with *Plp1-CreER^T2:ROSA26^tdTomato* mice. Accordingly, live-cell imaging must be performed to define the lifespan of a phagosome in normal utricles, which may differ from the lifespan of phagosomes after HC damage.

If we make a conservative estimate, based on SC fate-mapping alone, that 2 HCs are turned over per week, and we consider the mouse's lifespan to be 18 months, with an average of 4.3 weeks/month, then we can estimate that 155 HCs, or 4% of the HC population, are turned over during a mouse's lifetime. It seems unlikely that such a small degree of HC turnover would have a significant impact on vestibular homeostasis. However, we found evidence that type I HCs are cleared, but no evidence that type I HCs are replaced. A progressive loss of type I HCs over time or disruption in the balance of type II HC clearance and addition, for example during aging, could result in vestibular dysfunction. *Park et al. (1987)* measured a ~14% reduction in utricular HCs in old C57Bl/6NNia mice, and this reduction was similar for both type I and II HCs. By contrast, *Kirkegaard and Nyengaard (2005)* found no evidence of HC loss in aged outbred mice. Declines in vestibular function accompanying aging have been reported in some strains of mice (*Shiga et al., 2005*; *Mock et al., 2011*) and in humans (e.g., *Agrawal et al., 2009*), so it is important to further investigate if HC turnover is reduced with aging.

## Materials and methods

### Mouse models

Swiss Webster (stock #689), CBA/CaJ (stock #654), C57Bl/6J (stock #664), *Plp1-CreER^T2* (stock #5975; RRID:MGI:3696409; C57Bl/6J background; *Doerflinger et al., 2003*), ROSA26^CAG-loxP-stop-loxP-tdTomato (*ROSA26^tdTomato*)(also called Ai14, stock #7908; RRID:IMSR_JAX:007908; C57Bl/6J background; *Madisen et al., 2010*), and *Atoh1^GFP* (stock #13593; RRID:IMSR_JAX:013593; C57Bl/6J background; *Rose et al., 2009*) mice were purchased from The Jackson Laboratory (Bar Harbor, ME). *Lfng-eGFP* mice (RRID:MMRRC_015881-UCD, CD1 background) were generated by the GENSTAT program (*Gong et al., 2003*), were provided by Dr. Andrew Groves (Baylor College of Medicine, Waco, TX) for this study, and are available from Mutant Mouse Resource and Research Centers (strain name: B6;FVB-Tg[Lfng-EGFP]HM340Gsat/Mmucd). *Pou4f3^DTR* mice (C57Bl/6J or CBA/CaJ background; *Golub et al., 2012*) were provided by Dr. Edwin Rubel (University of Washington, Seattle, WA) and *Atoh1-CreER^TM* mice (RRID:MMRRC_029581-UNC; FVB/NJ background; *Chow et al., 2006*) were provided by Dr. Suzanne Baker (St. Jude Children's Research Hospital, Memphis, TN). Genotyping for *Plp1-CreER^T2*, *ROSA26^tdTomato*, *Atoh1^GFP*, and *Atoh1-CreER^TM* mice was performed by Transnetyx, Inc. (Cordova, TN). Genotyping for *Pou4f3^DTR* and *Lfng-eGFP* mice was described previously (*Golub et al., 2012*; *Burns et al., 2015*). Both genders were used in all studies. *Atoh1^GFP/GFP* mice carried two alleles (homozygous), and *Plp1-CreER^T2*, *ROSA26^tdTomato*, *Pou4f3^DTR*, *Atoh1-CreER^TM*, and *Lfng-eGFP* mice were heterozygotes. Crosses led to progeny with mixed strain background. All procedures were conducted in accordance with approved animal protocols from the

Institutional Animal Care and Use Committees at the University of Washington (Seattle, WA) and Southern Illinois University School of Medicine (Springfield, IL).

## Drug treatments

For fate-mapping and phagosome analyses in normal mice, tamoxifen [9 mg/40 g body weight, intraperitoneal injection (IP); Sigma-Aldrich (St. Louis, MO)] was injected once on two consecutive days in 6-week-old $Plp1$-$CreER^{T2}$:$ROSA26^{tdTomato}$ and $Atoh1$-$CreER^{TM}$:$ROSA26^{tdTomato}$ mice. For fate-mapping studies in damaged utricles, tamoxifen (6 mg/40 g body weight, IP) was injected once on two consecutive days in 9-week-old $Plp1$-$CreER^{T2}$:$ROSA26^{tdTomato}$:$Pou4f3^{DTR}$ mice, followed one week later by two intramuscular (IM) injections of DT. Each injection was 25 ng/g body weight, and there was a 48 hr interval between injections. DT was purchased from Sigma-Aldrich (St. Louis, MO) or List Biological Laboratories, Inc. (Campbell, CA). The different tamoxifen doses produced similar labeling of utricular SCs. Controls for normal utricles consisted of $Plp1$-$CreER^{T2}$:$ROSA26^{tdTomato}$ and $Atoh1$-$CreER^{TM}$: $ROSA26^{tdTomato}$ mice that did not receive tamoxifen injection and were housed separately from tamoxifen-treated mice. Controls for damaged utricles were littermates lacking the $Pou4f3^{DTR}$ allele ($Plp1$-$CreER^{T2}$:$ROSA26^{tdTomato}$ mice) that received both tamoxifen and DT injections.

For phagosome analyses in damaged utricles, DT was injected as described above, and mice were euthanized 4, 7, 14, 40, 90, or 120 days later. Controls were littermates lacking the $Pou4f3^{DTR}$ allele ($Plp1$-$CreER^{T2}$:$ROSA26^{tdTomato}$ mice) that received both tamoxifen and DT injections.

Methods for administering BrdU [Sigma-Aldrich (St. Louis, MO)] to adult mice are described in *Table 1*.

## Generation of semi-thin sections

Temporal bones were dissected, and a small hole in the temporal bone was generated adjacent to the oval window. Fixative (2% paraformaldehyde/3% glutaraldehyde in 0.1 M sodium phosphate buffer, pH 7.4) was injected into the hole, and temporal bones were immersion-fixed overnight at 4°C. Utricles were removed, and the otoconial membrane and otoconia were dissected away. Utricles were post-fixed in 1% osmium tetroxide and embedded in Eponate 12 Kit with DMP-30 (Ted Pella Inc., Redding, CA). Semi-thin sections (2 μm) were mounted onto gel-coated slides and stained with toluidine blue dye [Sigma-Aldrich (St. Louis, MO)]. Sections were imaged using a Zeiss Axioplan microscope (Zeiss, Jena, Germany).

## Immunofluorescent staining

Temporal bones were removed and either submerged in electron microscopy grade 4% paraformaldehyde (Polysciences, Inc., Warrington, PA) overnight at room temperature or partially dissected to remove otoconia from the utricle before fixation in 4% paraformaldehyde for two hours at room temperature. After fixation, temporal bones were stored in PBS until whole utricles were dissected out of temporal bones and placed in 96-well plates for free-floating immunofluorescent labeling. Utricles were washed twice in 1X PBS, then permeabilized with 2 mg/ml bovine serum albumin [BSA, Sigma-Aldrich (St. Louis, MO)] and 1.0% Triton X-100 (Sigma-Aldrich, St. Loius, MO) in 1X PBS for 1 hr at room temperature. Utricles were then incubated in blocking buffer [10% normal horse serum (Vector Laboratories, Burlingame, CA) and 0.4% Triton X-100 in 1X PBS] at room temperature for 3 hr. Primary antibodies were diluted in blocking buffer and incubated overnight at 4°C, except for the anti-BrdU, anti-GFP, and mouse anti-myosin VIIa antibodies that were incubated for 48–72 hr. The following primary antibodies were used: mouse anti-BrdU [1:300, RRID:AB_400326, BD Biosciences (San Jose, CA), #347580]; rabbit anti-calbindin [1:200, RRID:AB_2068336, EMD Millipore (Billerica, MA), #AB1778]; rabbit anti-calretinin [1:100, RRID:AB_2068506, EMD Millipore (Billerica, MA), #AB5054]; rabbit anti-aCasp3 [1:200, RRID:AB_397274, BD Biosciences (San Jose, CA), #559565]; rat anti-CD68 [1:100, RRID:AB_566872, AbD Serotec (Raleigh, NC), #MCA A341GA]; rabbit anti-espin [1:500, RRID:AB_2630385, gift from Dr. Stefan Heller, Stanford University]; rabbit anti-GFP [1:250, RRID:AB_221570, Invitrogen (Carlsbad, CA), #A6455]; rabbit anti-IBA1 [1:1000, RRID:AB_839505, Wako Pure Chemical Industries, LLC (Richmond, VA), #019–19741]; mouse anti-myosin VIIa [1:100, RRID:AB_2282417, Developmental Studies Hybridoma Bank (Iowa City, IA), #138–1]; rabbit anti-myosin VIIa [1:100, RRID:AB_2314839, Proteus Biosciences, Inc. (Ramona, CA), #25–6790]; rabbit

anti-PMCA2 [1:1000, RRID:AB_2630386, gift from Dr. Peter Barr-Gillespie, Oregon Health Sciences University]; mouse anti-POU4F3 [1:500, RRID:AB_2167543, Santa Cruz Biotechnology, Inc. (Dallas, TX), #sc-81980]; rabbit anti-PCDH15-CD2 [1:200, RRID:AB_2630387, gift from Drs. Tom Friedman (National Institutes of Health) and Zubair Ahmed (University of Maryland)]; goat anti-SOX2 [1:200, RRID:AB_2286684. Santa Cruz Biotechnology, Inc. (Dallas, TX), #sc-17320]; and rabbit anti-tenascin [1:100, RRID:AB_2256033, EMD Millipore (Billerica, MA), #AB19013]. After rinsing thrice with 1X PBS, utricles were incubated with Alexa Flour-conjugated secondary antibodies [1:400 in blocking buffer, Invitrogen (Carlsbad, CA)] for 2 hr at room temperature. Utricles were then incubated with Alexa Flour-conjugated phalloidin [1:100, RRID:AB_2620155 or RRID:AB_2315147, Invitrogen (Carlsbad, CA), #A12379 or #A22287] in 1X PBS for 30 min followed by a second 30 min incubation in DAPI (Sigma-Aldrich, St. Louis, MO) at 1 μg/ml in 1X PBS. After rinsing thrice in 1X PBS, whole utricles were mounted in Fluoromount-G (Southern Biotech, Birmingham, AL) between two cover slips anchored to a glass slide with putty.

To detect apoptosis in whole utricles and gut tissue (positive control), we utilized the ApopTag Fluorescein In Situ Apoptosis Detection Kit [#S7110, Chemicon (now EMD Millipore, Billerica, MA)] according to the manufacturer's instructions followed by immunostaining as described above.

To detect proliferating cells in whole utricles and gut tissue (positive control) from mice treated with BrdU, fixed tissue was first labeled with antibodies for cell specific markers as described above. After secondary antibody incubation, tissue was fixed again in 4% paraformaldehyde for 20 min at room temperature. After rinsing in 1X PBS and a second permeabilization step, the tissue was incubated in 2 N HCl in 0.05% Triton X-100 for 1 hr at room temperature, then rinsed in 1X PBS, and placed in blocking buffer for 30 min. Then the tissue was immunostained with anti-BrdU primary antibodies as described above.

## Confocal microscopy and image analysis

Fluorescent images were obtained with an Olympus FV-1000 microscope (Olympus, Center Valley, PA). In most cases, z-series images were collected with a 60x oil objective through the entire macula, from above stereocilia bundles to the stroma below the basal lamina, at 0.5 or 0.25 μm increments. For qualitative analyses, we generated z-series in 2–3 extrastriolar and striolar regions. For quantitative analysis for *Figures 3*, *5*, *6*, *7*, *8* and *9*, we generated high-resolution images or montages of the entire utricular macula. Image analysis was performed using Fiji (http://fiji.sc/). Fiji's Cell Counter plugin was used for quantitative analyses.

To determine the percentage of SCs labeled in *Plp1-CreER$^{T2}$:ROSA26$^{tdTomato}$* mice one week after tamoxifen injection, eight 50 × 50 μm regions of each utricle (3 in the medial extrastriola, 3 in the lateral extrastriola, and two in the striolar area) were sampled. All SCs within the sampled regions were counted and scored as tdTomato-positive or tdTomato-negative. The average percentage of extrastriolar SCs that were tdTomato-labeled was calculated by combining medial and lateral extrastriolar regions.

To assess whether phalloidin-labeled phagosomes co-localized with IBA1 immunolabeling, 301 phalloidin-labeled phagosomes from 6 Swiss Webster mice at 9 weeks of age were closely examined. To quantify PCDH15-CD2-labeled stereocilia bundles in adult CBA/CaJ and Swiss Webster mice, the 100x oil objective on the Olympus FV-1000 confocal microscope was used. The observer looked over the entire utricle and counted phalloidin-labeled stereocilia bundles co-localized with multiple puncta of PCDH15-CD2 along the length and width of the bundle.

## Criteria for cell typing

HCs were distinguished from SCs by the presence of a HC marker and the location of their nuclei, which reside above the clear monolayer of SC nuclei along the basal lamina (*Figure 1D*). We used several morphological criteria to classify HCs as type I or type II in whole-mounted utricles. These criteria for mice are defined in many publications, including *Rüsch et al. (1998)* and *Pujol et al. (2014)*, are described in *Figure 1D*, and shown in *Videos 4* and *5*. Almost all morphological analyses were performed in whole-mount utricles labeled with antibodies to myosin VIIa, which label the cytoplasm of both type I and type II HCs. We acquired z-series images through the entire sensory epithelium (from the lumen through the basal lamina) in striolar and extrastriolar regions. Images were analyzed thoroughly off-line by scanning up and down through the z-series, examining the entire

body of each HC. This approach allowed us to assess all criteria in each image and compare features amongst several cells. Four morphological criteria were used to classify HCs across all regions of the utricle: thickness of the apical cytoplasm, thickness of the perinuclear cytoplasm, size and shape of the nucleus, and presence/absence of basolateral processes. The apical cytoplasm of type I HCs constricts significantly below the cuticular plate, while the apical cytoplasm of type II HCs does not. The perinuclear cytoplasm is very thin in type I HCs and considerably thicker in type II HCs. The nuclei in type I HCs are smaller and round, while the nuclei of type II HCs are larger and oblong. The clearest distinguishing features between type I and II HCs throughout the entire utricle (striola and extrastriola) are basolateral processes (*Pujol et al., 2014*), which are cytoplasmic portions of type II HCs that extend below the nucleus and are fully labeled by anti-myosin VIIa antibodies. In the extrastriola (the largest area in the utricle), there is a fifth powerful criterion for HC typing: the nuclei of type I and II HCs are located in distinct near-monolayers. Type II HC nuclei are consistently located apical to type I HC nuclei (*Videos 4* and *5*). In the striola, where type II and type I HC nuclei are less clearly stratified, we used the 4 criteria listed above—thickness of the apical and perinuclear cytoplasm, nuclear size and shape, and presence or absence of a basolateral process. In studies with *Plp1-CreER^{T2}:ROSA26^{tdTomato}* and *Plp1-CreER^{T2}:ROSA26^{tdTomato}:Pou4f3^{DTR}* mice, the cytoplasm of most SCs was labeled with tdTomato. As a result, we had another criterion to help us with typing: type II HCs were completely encompassed by the tdTomato-positive SC processes, while type I HCs had an unlabeled gap around them, which was the calyx. Any cells that did not clearly match criteria described above for any analysis were scored as an 'unknown.'

## Constructing utricle maps

The Cell Counter plugin in Fiji was used to mark tdTomato-labeled HCs in one utricle from an adult *Plp1-CreER^{T2}:ROSA26^{tdTomato}* mouse (*Figure 6F*). Results were imported into Microsoft Powerpoint 2011 (Microsoft, Redmond, WA) to assemble the images and reconstruct an entire utricle. The file was imported into Adobe Photoshop CS 4 (Adobe, San Jose, CA), where an outline was drawn around the whole utricle and dots were drawn directly over the Cell Counter labels to create the maps of tdTomato-labeled HCs.

To generate the map of PCDH15-CD2-labeled bundles (*Figure 5C*), an adult Swiss Webster utricle was scanned at high magnification using confocal microscopy, and the position of each PCDH15-CD2-labeled bundle was noted for each field on a grid. The utricle was reconstructed using Adobe Photoshop CS 4.

## Statistics

Data were analyzed using Graphpad Prism 5.0 (Graphpad Software, La Jolla, CA) or SAS 9.4 (Cary, NC). All data are reported as mean ±1 standard deviation. 95% confidence intervals were calculated with Microsoft Excel 2011 (Microsoft, Redmond, WA). For all studies, one utricle per animal was assessed and the 'n' value represents the number of mice included in the study, unless otherwise noted. n's for each data set are included in the Results section, source data tables, and/or the Figure Legends.

## Acknowledgements

This work was supported by the National Institutes of Health (F32 DC013695 to SAB, R01 DC013771 to JSS, and P30 DC04661 to the UW Research Core Center) and the Office of Naval Research (N00014-13-1-0569 to BCC). For technical assistance, we thank Glen MacDonald, Jialin Shang, Irina Omelchenko, Linda Robinson, Brandon Warren, Connor Finkbeiner, Andrea Gavrilescu, and Caitlyn Trullinger-Dwyer from the University of Washington and Michelle Randle and Kaley Graves from Southern Illinois University School of Medicine. We thank Dr. Kristin Delfino (Southern Illinois University School of Medicine) for assistance with statistical analyses. We are grateful to Dr. Patricia White (University of Rochester) for sharing transgenic mice with the *Atoh1* 3' enhancer sequence driving GFP expression (*Lumpkin et al., 2003*), used in our early experiments and are not discussed here, but provided early clues to the existence of utricular HC addition in normal adult mice. We thank the following colleagues for their assistance: Dr. Andrew Groves (Baylor College of Medicine) for helpful discussions and for providing *Lfng-eGFP* mice and *Atoh1^{GFP}* mouse tissue; Dr. Peter Barr-Gillespie (Oregon Health Sciences University) for sharing PMCA2 antibodies; Drs. Thomas Friedman (National

Institutes of Health) and Zubair Ahmed (University of Maryland) for sharing PDCH15-CD2 antibodies, Dr. Stefan Heller (Stanford University) for sharing espin antibodies. We are grateful to Dr. Suzanne Baker (St. Jude Children's Research Hospital) for sharing *Atoh1-CreER*<sup>TM</sup> mice. We thank Dr. Edwin Rubel (University of Washington) for sharing *Pou4f3*<sup>DTR</sup> mice and many helpful discussions. Finally, we are very appreciative of Drs. David Raible (University of Washington) and Mark Warchol (Washington University) for discussion and comments on the manuscript.

## Additional information

### Funding

| Funder | Grant reference number | Author |
|---|---|---|
| National Institutes of Health | F32 DC013695 | Stephanie A Bucks |
| National Institutes of Health | DC013771 | Jennifer Stone |
| National Institutes of Health | DC04661 | Jennifer Stone |
| Office of Naval Research | N00014-13-1-0569 | Brandon C Cox |

The funders had no role in study design, data collection and interpretation, or the decision to submit the work for publication.

### Author contributions

SAB, Conceptualization, Data curation, Formal analysis, Methodology, Writing—original draft, Writing—review and editing; BCC, Conceptualization, Investigation, Methodology, Writing—original draft, Writing—review and editing; BAV, JPM, Formal analysis; TBN, Formal analysis, Methodology; JSS, Conceptualization, Data curation, Formal analysis, Supervision, Funding acquisition, Investigation, Methodology, Writing—original draft, Project administration, Writing—review and editing

### Author ORCIDs

Jennifer S Stone, http://orcid.org/0000-0002-6742-0590

### Ethics

Animal experimentation: All procedures were conducted in strict accordance with recommendations in the Guide for the Care and Use of Laboratory Animals of the National Institutes of Health. All animals were handled according to approved institutional animal care and use committee (IACUC) protocols at the University of Washington and Southern Illinois University School of Medicine, which are fully accredited (Animal Welfare Assurance numbers A3464-01 and A-3209-01). Every effort was made to minimize suffering.

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
