## [Decision Letter]

Thank you for submitting your article "Supporting cells remove and replace sensory receptor hair cells in a balance organ of adult mice" for consideration by *eLife*. Your article has been reviewed by 3 peer reviewers, and the evaluation has been overseen by a Tanya Whitfield as the Reviewing Editor and Sean Morrison as the Senior Editor. The following individuals involved in review of your submission have agreed to reveal their identity: Yehoash Raphael (Reviewer #2) and Jonathan Gale (Reviewer #3).

The reviewers have discussed the reviews with one another and the Reviewing Editor has drafted this decision to help you prepare a revised submission.

This paper demonstrates that a slow endogenous turnover of vestibular hair cells in the utricle of adult mice occurs in the absence of overt hair cell damage, via phagocytosis, apparently by supporting cells. The paper also demonstrates that the mammalian vestibular system can regenerate Type II vestibular hair cells through phenotypic conversion of supporting cells.

Overall, it was agreed that the study communicated a number of significant and important new findings and would be of general interest. However, there were several concerns that should be addressed before publication. The following numbered points were felt to be the essential changes required; the full reviews are appended below for further information.

1) Two of the reviewers query whether the rate of hair cell turnover in the Swiss-Webster mice is specific to that strain. Further information and discussion is required here.

2) Given the known phagocytic activity of Schwann cells, and the position and labelling of Schwann cells in the utricle, please provide further data, if possible, to address the possibility that the phagosomes are Schwann cell-derived (see comments from reviewer 1). As a minimum, further discussion is required to address this point. At present, Schwann cells are discussed as a possible source of hair cells, but not of phagosomes, in the manuscript.

3) As requested by reviewer 2, please remove the word 'undamaged' when referring to normal turnover of hair cells and discuss the possibility that such turnover may replace cells that are damaged in some way by normal wear and tear.

4) As requested by reviewer 2, please provide further data and discussion to compare the role of supporting cells in clearing damaged hair cells after diphtheria toxin lesion with their role in normal turnover.

5) Please review the statistical analysis of the data carefully (see comments from reviewer 2).

6) It was felt that the last paragraph of the Discussion makes an abrupt end to the manuscript and does not tie the story together well. Please aim to rewrite this, in particular to include a clear discussion of the physiological significance of the findings.

7) After discussion, it was felt that answering the interesting questions raised by reviewer 1 concerning mechanistic insights would be beyond the scope of this study; therefore, it is not essential to address these with further experimental work. However, if the authors have further discussion to add here, this would help to address these concerns.

*Reviewer #1:*

The manuscript by Bucks et al. explores important issues, i.e. the ability of the mammalian inner ear to replace hair cells and the mechanisms by which this might occur. The study reports that hair cells in the adult mammalian utricle undergo removal and replacement by their surrounding supporting cells under both normal and damaging condition. The authors performed thorough analysis characterizing three individual mouse strains and using lineage tracing experiments to confirm the cellular origin of replaced hair cells. The manuscript is clear and technically sound. However, there are major issues regarding to the general interest of this study and the physiological significance of the findings.

Major comments:

Novelty: the paper refers to previous studies that showed the ability of adult rodents to regenerate small numbers of utricular HCs after ototoxin-induced damage. Was the magnitude of the regeneration in those studies much lower than in the current study? Since in this paper the authors show that the degree of cell replacement differs between mouse strains, are the present results more dramatic because the specific strain studies here? The authors need to discuss if the high degree of cell replacement seen in Swiss Webster mice is an anomaly given that replacement occurs at a much lower degree in the other strains.

The paper also refers to previous studies showing that supporting cells phagocytose hair cells after ototoxic damage in and mammals, and there have been reports of vestibular hair cell loss in the adult ear. Thus, it seems that this study just provides further evidence of the hair cell phagocytic capacity of supporting cells in the normal tissue.

Biological-physiological relevance: Although this study provides solid evidence of hair cell replacement in adult utricle, primarily by transdifferentiation of supporting cells, it remains unclear if 36.6 cells replaced per utricle at 15 wk post tamoxifen (Figure 5), which would translate into about 317 in 2.5 years, the lifespan of a mouse is biologically significant. This number of hair cells, if I'm not mistaken, represents a very small% of total hair cells in the utricle, making me wonder what is the physiological relevance of these findings. Furthermore, the number of cells replaced is much lower in several strains, further questioning the functional impact of such process. Does this percentage increase over a longer period of time? Are these tdTomato-traced cells functional? The replaced hair cells typically have shorter stereocilia bundles, how does this affect hair cell function? Do the replaced hair cells form ribbon synapses with vestibular ganglion terminals?

Mechanistic insights: Conversion of supporting cells to hair cells appears to be a critical pathway for hair cell regeneration, however, the mechanistic details are lacking for this study to be sufficiently novel and of general importance. For example, what is the signal that triggers the supporting cell transdifferentiation? How does loss of supporting cells (by conversion to hair cells) affect vestibular function? How are the vestibular ganglion processes remodeled during hair cells loss and replacement?

Other issues:

In the 6th paragraph of the subsection “Type I and II HCs are cleared by phagosomes from adult mouse utricles under normal conditions”: the authors report that in 5-10 week-old Swiss Webster mice, 71.7% of the on average 48 phagosomes per utricle are not hair cells, leaving the reader wondering what is the most intense phagosome activity in the utricle related to?

The paper shows SCs phagocytose both type I and II HCs, but only generate type II HC. Does the number of type I HCs decrease with age in the undamaged utricle?

Although several morphological criteria are used to classify HCs as type I or type II in utricle whole-mounts, use of specific antibodies, as in Figure 1, would be necessary to make a compelling case.

As shown by the authors, in *Plp-CreERT2:ROSA26^tdTomato^* mice, some Schwann cells in the stroma and cells in the region of transitional epithelium are also tdTomato-positive. Since F-actin enriched phagosomes are found in the SC nuclear layer, which is adjacent to the basal lamina and stroma, it could be a possibility that phagosomes are derived from Schwann cells. And phagosomes in the region near the transitional epithelium (as shown in Figure 2) may be derived from cells in the transitional epithelium. These possibilities should be considered in the Discussion.

Figure 4: it would be helpful is the authors would explain what is the PCDH15-CD12 signal in the adult (Figure 4).

The last paragraph of the Discussion section does not flow with the rest of the text, seems misplaced.

*Reviewer #2:*

This is an elegant paper reporting findings related to hair cell elimination and spontaneous regeneration in the vestibular epithelium.

"In mammals, vestibular HC production is reported to occur only during gestation and the first two postnatal weeks…”

This timing may be correct for mice but not necessarily all mammals. Rephrase please.

"Supporting cells (SCs) are glia-like cells". Also: "SCs, which resemble glia".

SCs are epithelial cells. True, they have some functions and some molecular properties shared with glia, but the definition "glial-like" may not be accurate.

Previous studies, including from the Seattle group, claimed that SCs are ejected from the sensory epithelium when eliminated by an insult. How do the current results conform to these findings? Please discuss to explain this.

When SCs clear HCs after ototoxin-mediated injury, they first produce an actin cable to constrict the HC. This was shown by Meiteles in 1994 (PMID: 7528737). Add citation.

Assessment of the role of SCs in the death of HCs is presented in details as related to turnover, but not described for the DT lesion in the DTR mice, where it is even more important. Add data and discuss differences between turnover and response to trauma.

Figure 1 should indicate surface view more clearly. Could dispense with this image because all micrographs are at supporting cell level. A schematic at the level of SCs, would be more useful to add or replace.

Figure 1 legend: "In most panels…" should state here how the other panels are labeled.

The actin rod was first described by Sobin and later termed cytocaud by Kanzaki upon further characterization. Better citations would help. Cytocauds are compatible with survival of HCs for weeks or longer, yet they are clearly manifestations of stress and onset of degeneration. Therefore, their presence is not necessarily related to the mode of HC elimination.

Figure 1 is the merge? There seems to be more myosin at this level than in just the 4 areas attributed to ectopic HCs.

What is "brightest point projection image"?

Addition of a neuronal marker for calyces would help reader to distinguish type I from type II HCs.

What is "basket-shape"? Baskets come in wide variety of shapes. A more specific geometric term should be used.

It was not clear what the background of *plp cre* was, and whether Swiss Webster, CBA or c57b6 was the closest comparison.

I would suggest not using the term "undamaged". An agent may not have been used to intentionally induce a lesion, but one cannot rule out the possibility that the cells targeted for phagocytosis are not damaged. It may be normal wear and tear but it is still damage.

In the second paragraph of the subsection “Immature HCs are present in undamaged utricles of adult mice”: larger format images similar to Figure 4 should be used to show the difference between ages in localization pcdh15 isoforms across utricular regions. This would be better support for the comparison to the distribution of atoh1.

Figure 5 and Figure 6 – aren't these addressing the same or at least overlapping questions? Type I HCs don't increase so they are not being generated by either proposed mechanism.

In the last paragraph of the subsection “*Atoh1-CreER^TM^*-labeled type II HCs did not convert into type I HCs under normal conditions”: Statistics on proportion is problematic because proportions are ratios, which are not normally distributed. They also do not address the potential interdependence of the numerator and denominator. A test for correlation between counts would be better – i.e. does slope differ between time points. Also, if the numerator and denominator are not truly dependent and independent, you might consider reduced major axis regression. It also seems like the data in Figure 5 might suffice to answer this.

No need to include both sd and ci; one can be calculated from the other.

Sample sizes are too small to justify a test assuming unequal variances; it may seem obvious that 11x difference in variances would be significant, but these sample sizes are too small for credible comparisons of variances, and it is not possible to know if they represent normal distributions.

*Reviewer #3:*

In this paper the authors investigate the mechanisms of sensory hair cell turnover in the mammalian vestibular system. Understanding these mechanisms is important if we are to fully understand how to regenerate sensory hair cells in humans.

Firstly, the authors describe finding actin-rich structures that are closely associated with hair-bundle less hair cells underneath the utricular epithelial surface close to the basal lamina. These structures were found in fixed specimens of control mice (of various strains). There are a small number of these in a single sense organ at any one time (ranging from 10 to 60 depending on the strain but independent upon the age of the mouse). They are somewhat regionally restricted, being found predominantly in the peri-striolar region.

The authors suggest that the structures are similar to the supporting cell based phagosomes that have been described in both chicken and murine utricles after ototoxin-induced hair cell death. If so they would be derived from supporting cells rather than another cell type and the authors use a *Plp-CreERT2:ROSA26^tdTom^*mouse to show that the actin -rich structures that they have discovered are derive from a supporting cell. The authors suggest that these structures indicate a homeostatic(?) removal process for hair cells during the "normal" lifespan of a mouse and so predict that there should be immature cells that need to be added in order to maintain cell numbers. They find that adult mouse utricles do have hair cells with "immature" characteristics, as has been suggested by other work, and then use an Atoh1-GFP mouse to provide support for this finding (since we know that with maturation AtoH1 expression is significantly reduced as hair cells mature). Furthermore, they use the PlpCRE reporter mice to determine the number of hair cells that are derived from Plp-expressing cells and find that this number increases over a period of 15 weeks after Cre activation – an addition of ~30 cells over 15 weeks. Detailed analysis revealed that it was almost exclusively the Type II hair cell that was added during this period, and using another mouse model they show that the Type II hair cells do not convert to the other Type I hair cells, at least over the 15 week period monitored. Finally, by crossing their PlP reporter mice with a Pou4f3-Diphtheria Toxin mouse (that they have described previously) the authors showed that the ablation of hair cells resulted in a substantial increase (6x) in the numbers of tdTom-positive hair cells i.e. hair cells derived from SCs

General Comments:

The paper is a great example of what can be done using different mouse models and different cellular markers. The work is high quality and is well presented. The paper gives us an important insight into the life and death of hair cells in the mammalian utricle under "normal" conditions and shows the importance of the supporting cells in that life and death.

Specific comments:

1) The data presented suggest during normal ageing a small number of hair cells are removed from the epithelial surface and become associated with the actin-rich phagosomal structures described in the paper. It is not clear to me from the data presented whether the phagocytic process that is suggested in Figure 1 and Figure 2 is any different from the mechanisms that have been described and shown directly through live-imaging in both the avian and mammalian utricle. It seems very possible that the same process that has been described for ototoxically damage cells could be responsible for the removal of the cells described here. Is there any reason not to conclude that a similar process is used? This points warrants further discussion (note – see point 12 below)?

2) There is a description of "association" between phagosomes and hair cells e.g. subsection “Type I and II HCs are cleared by phagosomes from adult mouse utricles under normal conditions”, third paragraph, fourth and sixth paragraphs. A qualification of what constituted such "associations" and how there were determined by the observer is required. Does this association mean that there was clear evidence of engulfment of a hair cell within a phagosome?

3) In order to call this structure a phagosome there needs to be better description of how the actin rich structures that are shown are linked to the supporting cells that presumably generated them – this is hard to see this from the images provided. Have the authors for example tried to reconstruct the phagosome structure in 3D from the confocal stacks they have?

4) Is there something unusual about Swiss Webster mice? They have ~3x as many of the actin rich structures as the other two strains?

5) The phagosome-like structures are somewhat regionally restricted, being found predominantly in the peri-striolar region. There is little discussion of why this might be the case.

6) Can the authors estimate the lifetime of the actin-rich phagosomal structures they describe (see note 12 below)?

7) How much actin is there in these structures compared to the actin-rich hair bundles of the normal hair cells? How about the spikes – the sequence showing the stack (Video 2) suggests very similar levels?

8) How are the estimations for the% of Type I and Type II hair cells made (subsection “*Atoh1-CreER^TM^*-labeled type II HCs did not convert into type I HCs under normal conditions”, first paragraph)? I estimated 3% and 35% respectively using the numbers of tdTom labelled hair cells shown in the table for 7 weeks of age, a total hair cell number of 3600 and a ratio of 1.17:1.

9) At the end of the subsection “Diphtheria toxin-induced HC ablation increased SC-to-type II HC transition in adult mouse utricles” – it is stated that the rate of transdifferentiation is significantly increased. The data indicates that this is what happens but no rates are provided per se. Consider revising the sentence?

10) The finding that only Type II hair cells are replaced in the mammal has implications for the recovery of function since the Type I cells provide/send different information to brain. Does this finding concur with what is known/has been shown for the functional recovery that has been described in other work?

11) The authors show in fate mapping experiments that ~2 hair cells are added per week. If we assume that the utricle maintains its normal number of cells over this period (which is thought to be the case) then the rate of removal should be the same. Even if the fate-mapping studies are not 100% effective and we assume a 75% efficacy then the simplest estimation is that there would be~3 (or just under) hair cells turned over per week, rather than the 10 suggested at the end of the subsection “Estimates of the rate of HCs turnover in adult mice”. Using this estimate we can see that the phagocytic structures that are described here is a somewhat long lived with lifetimes >1 week, indicating some differences to those phagosomes already described by others.

Other comments:i) Subsection “Type I and II HCs are cleared by phagosomes from adult mouse utricles under normal conditions”, fourth paragraph: How do the authors know that this is a calyceal afferent from the image shown in Panel 1F? This is quite hard to see and a marker would be required for definitive evidence. The tenascin data shown in Figure 1 provide some support for this although it is not entirely clear that the tenascin is within the actin-rich structure. Again – could some 3D reconstruction help?

ii) Fig1L – it is hard to make out where the epithelial surface is perhaps this can be clarified.

iii) Figure 2 – clarify that this is a projection of the lower planes of the utricle – if it was a max projection of the full stack you would see mostly hair bundles in the image wouldn't you?

iv) Figure 4 – is the edge of the sensory macula in the field of view or is it an oblique optical section in both cases?

v) I found Figure 5 hard to see due to the merging of the colours. It is clear that in C' there is a tdTom-positive and Myo7a-positive double labelled cell. Why does this cell have such a strong Myo7a signal – is this typical?

---

## [Author Response]

*[…] Overall, it was agreed that the study communicated a number of significant and important new findings and would be of general interest. However, there were several concerns that should be addressed before publication. The following numbered points were felt to be the essential changes required; the full reviews are appended below for further information.*

*1) Two of the reviewers query whether the rate of hair cell turnover in the Swiss-Webster mice is specific to that strain. Further information and discussion is required here.*

This is an excellent question. After conducting an extensive literature search, the only notable Swiss Webster phenotype we found was two reports describing Swiss Webster mice exhibiting circling behavior and inflammation in vestibular organs caused by infection with *Pseudomonas aeruginosa* (Ediger et al., 1971 Lab Anim Sci; Olson and Ediger, 1972 Lab Anim Sci). We have not observed circling behavior in our Swiss Webster mice, and no evidence of vestibular macular inflammation or damage was detected during our analyses of their utricles.

To date, there is very little known to form a clear hypothesis as to why Swiss Webster mice have seemingly increased HC turnover compared to other strains in our study. It is important to note, however, that while our findings describe significantly more phagosomes in Swiss Webster mouse utricles, there was no significant increase in protocadherin15-CD2 labeled stereocilia bundles compared to other strains. Assuming that there is a balance between HC clearance and HC addition, the actin dynamics and, consequently, kinetics of creating and resolving phagosomes may be longer in Swiss Webster mice. Live-imaging studies of HC turnover in utricles from Swiss Webster and other mouse strains could help clarify this. We have added the several sentences to the Discussion for this point (subsection “Significance of vestibular HC turnover in adult mice”, second paragraph).

*2) Given the known phagocytic activity of Schwann cells, and the position and labelling of Schwann cells in the utricle, please provide further data, if possible, to address the possibility that the phagosomes are Schwann cell-derived (see comments from reviewer 1). As a minimum, further discussion is required to address this point. At present, Schwann cells are discussed as a possible source of hair cells, but not of phagosomes, in the manuscript.*

This is an important question. Unfortunately, despite considerable effort, we were not able to find antibodies that label Schwann cells but not SCs or evidence of CreER mouse lines that target Schwann cells specifically. This is also true for cells in the transitional epithelium (TE). Thus, new tools are needed to delineate what, if any, contribution Schwann cells or TE cells make to the production of phagosomes and the addition of HCs in the normal utricle. However, we have included new data to further support the interpretation that many phagosomes derive from SCs. The *Lfng-eGFP* mouse expresses eGFP specifically in SCs in vestibular epithelia, but not in Schwann cells or TE cells (see revised Figure 4 and Burns et al., 2015 Nat Comm). In *Lfng-eGFP* mice, eGFP is expressed in the majority of SCs, and we observed co-localization of eGFP with F-actin rich phagosomes. This supports our findings using *Plp1-CreER^T2^:ROSA26^tdTomato^*mice that SCs can act as phagosomes. We have included these new data in Figure 4 and in Results (subsection “SCs clear HCs from adult mouse utricles under normal conditions”, second paragraph) and Materials and methods (subsection “Mouse models"). In addition, we have expanded the Discussion to include the possibility that Schwann cells could also act as phagocytes and that new tools are needed to investigate this idea (subsection “Phagocytosis by SCs mediates HC clearance in normal mouse utricles”, first paragraph). However, it is unlikely that TE cells are sources of phagosomes, since the majority of phagosomes we observed were located in the central region of the macula, away from the transitional epithelium. This point was also added to the Discussion (see the aforementioned paragraph).

*3) As requested by reviewer 2, please remove the word 'undamaged' when referring to normal turnover of hair cells and discuss the possibility that such turnover may replace cells that are damaged in some way by normal wear and tear.*

Thank you for this suggestion, which clarifies the study design for the reader. We replaced the term “undamaged” with “normal” throughout the text, including section headings and figure legends. For discussion of the *Plp1-CreER^T2^:ROSA26^tdTomato^:Pou4f3^DTR^* mouse experiments, the term “undamaged” was replaced with “control”. We also added a sentence to the Discussion to acknowledge that normal wear and tear may cause HC damage and induce phagocytosis (subsection “Phagocytosis by SCs mediates HC clearance in normal mouse utricles”, last paragraph).

*4) As requested by reviewer 2, please provide further data and discussion to compare the role of supporting cells in clearing damaged hair cells after diphtheria toxin lesion with their role in normal turnover.*

We counted phagosomes in *Pou4f3^DTR^* mice and created a new figure (Figure 9), which shows a ~60% increase in the number of phagosomes at 4 and 7 days post DT-induced damage, but not at other time points post DT. Our previous work showed a loss of >2000 HCs over the first week post-DT (Golub et al., 2012), so the increase of phagosomes we observed (~5 per utricle) in this time is much less than the thousands of HCs that need to be cleared. Kaur et al. (2015) described macrophage infiltration in DT-treated mouse utricles to clear HC debris, and these macrophages did not overlap with the F-actin-rich phagosome structures that we attribute to SCs. Thus, it is likely that SCs form phagosomes for normal HC turnover and minimally increase their phagocytic activity after damage. We also found that phagosomes in *Pou4f3^DTR^* mice were located in the normal HC layer, which contrasts with the basal location of phagosomes in normal utricles in the SC nuclear layer. In addition, some HCs in *Pou4f3^DTR^* mice co-localized with TUNEL or had apoptotic features in their nuclei, but these cells with apoptotic features were not associated with phagosomes. This contrasts with normal utricles, in which HCs associated with phagosomes had normal chromatin and no signs of apoptosis. We added these points to the Results (subsection “SCs play a minor role in the clearance of dying HCs after DT-induced damage”) and created a new Figure 9. In the Discussion, we compared phagocytosis between normal and HC damaged conditions (subsection “Phagocytosis by SCs mediates HC clearance in normal mouse utricles”, second and third paragraphs).

*5) Please review the statistical analysis of the data carefully (see comments from reviewer 2).*

We consulted with a statistician and made changes to the statistical analyses in question. A specific response to these comments from reviewer 2 is presented below (see our seventeenth response to review 2).

*6) It was felt that the last paragraph of the Discussion makes an abrupt end to the manuscript and does not tie the story together well. Please aim to rewrite this, in particular to include a clear discussion of the physiological significance of the findings.*

We have rewritten the last 3 paragraphs of the Discussion to address multiple reviewer comments. In addition to making estimates for how many HCs are turned over in a mouse’s lifetime, we conclude with a discussion for the possible functional significance of our findings. Specifically, we found evidence that type I HC are cleared, but no evidence that type I HCs are replaced. A progressive loss of type I HCs over time or disruption in the balance of type II HC clearance and addition, for example during aging, could result in vestibular dysfunction and balance disorders such as vertigo. Previous studies on HC numbers in aged mouse utricles and vestibular dysfunction during aging were also discussed (subsection “Significance of vestibular HC turnover in adult mice”).

*7) After discussion, it was felt that answering the interesting questions raised by reviewer 1 concerning mechanistic insights would be beyond the scope of this study; therefore, it is not essential to address these with further experimental work. However, if the authors have further discussion to add here, this would help to address these concerns.*

We agree that the mechanism of HC clearance and addition is very important to understand and are currently designing studies to address these questions. In the Discussion, we expanded the paragraph focused on why HCs are targeted for clearance suggesting damage caused by normal “wear and tear,” changes in innervation or cell membrane integrity, or even that SCs act as primary phagocytes (subsection “Phagocytosis by SCs mediates HC clearance in normal mouse utricles”, last paragraph). We also added a new paragraph to compare the mechanism of phagocytosis in normal versus HC damaged utricles as well as the cell types involved in each circumstance (subsection “Phagocytosis by SCs mediates HC clearance in normal mouse utricles”, third paragraph).

*Reviewer #1:*

*The manuscript by Bucks et al. explores important issues, i.e. the ability of the mammalian inner ear to replace hair cells and the mechanisms by which this might occur. The study reports that hair cells in the adult mammalian utricle undergo removal and replacement by their surrounding supporting cells under both normal and damaging condition. The authors performed thorough analysis characterizing three individual mouse strains and using lineage tracing experiments to confirm the cellular origin of replaced hair cells. The manuscript is clear and technically sound. However, there are major issues regarding to the general interest of this study and the physiological significance of the findings.*

*Major comments:*

*Novelty: the paper refers to previous studies that showed the ability of adult rodents to regenerate small numbers of utricular HCs after ototoxin-induced damage. Was the magnitude of the regeneration in those studies much lower than in the current study? Since in this paper the authors show that the degree of cell replacement differs between mouse strains, are the present results more dramatic because the specific strain studies here? The authors need to discuss if the high degree of cell replacement seen in Swiss Webster mice is an anomaly given that replacement occurs at a much lower degree in the other strains.*

We believe that there is confusion here between HC regeneration (which occurs after HC damage) and HC addition (which occurs under normal, physiological conditions). Previous studies have quantified HC regeneration in vestibular organs, but our manuscript is the first to definitively demonstrate and quantify HC addition in normal, adult utricles. While comparison of these quantifications is informative, they occur in two distinct environments and are not expected to result in the same number of added HCs. Also in our Figure 8, which quantifies the amount of HC regeneration in *Pou4f3^DTR^* mice, we only counted the tdTomato-positive HCs fate-mapped by *Plp1-CreER^T2^:ROSA26^tdTomato^*mice. This is likely an underestimate of total HC regeneration reported in previous studies since not all SCs are labeled by tdTomato in these mice.

For Swiss Webster mice, our findings describe significantly more phagosomes for HC clearance, but there was no significant difference in protocadherin15-CD2 labeled stereocilia bundles (labeling immature HCs) compared to another strain. Thus, we do not have evidence that there is an increase in HC addition in Swiss Webster mice. As explained in Essential Comment #1, we expanded the Discussion to address Swiss Webster’s relatively high numbers of phagosomes (subsection “Significance of vestibular HC turnover in adult mice”, second paragraph).

*The paper also refers to previous studies showing that supporting cells phagocytose hair cells after ototoxic damage in and mammals, and there have been reports of vestibular hair cell loss in the adult ear. Thus, it seems that this study just provides further evidence of the hair cell phagocytic capacity of supporting cells in the normal tissue.*

While several studies have demonstrated SCs act as phagocytes after HC are damaged (by noise, ototoxins, etc.), there are no previous studies which demonstrate that SCs phagocytose HCs in normal utricles. In addition, the reports of HC loss are from the aged ear and our study is focused on young adult mice. There are no reports of HC loss in utricles of young normal mice. Therefore, our findings suggest a previously unknown role of SCs as phagocytes of type I and II vestibular HCs under normal conditions.

*Biological-physiological relevance: Although this study provides solid evidence of hair cell replacement in adult utricle, primarily by transdifferentiation of supporting cells, it remains unclear if 36.6 cells replaced per utricle at 15 wk post tamoxifen (Figure 5), which would translate into about 317 in 2.5 years, the lifespan of a mouse is biologically significant. This number of hair cells, if I'm not mistaken, represents a very small% of total hair cells in the utricle, making me wonder what is the physiological relevance of these findings. Furthermore, the number of cells replaced is much lower in several strains, further questioning the functional impact of such process. Does this percentage increase over a longer period of time? Are these tdTomato-traced cells functional? The replaced hair cells typically have shorter stereocilia bundles, how does this affect hair cell function? Do the replaced hair cells form ribbon synapses with vestibular ganglion terminals?*

These are excellent points. We modified the end of the Discussion to address the functional significance of our findings (see Essential Comment #6). In addition, we explain in the Discussion (subsection “Significance of vestibular HC turnover in adult mice”, first paragraph) that our quantification of HC addition is likely an underestimate since not all SCs were fate-mapped and other cells (such as cells in the transitional epithelium) may also convert into HCs in the normal utricle.

As explained in our first response to reviewer 1, we do not have evidence that there is an increase in HC addition in Swiss Webster mice. Our data only show an increase in the number of phagosomes in Swiss Webster mice, which could be caused by strain-differences in actin dynamics and lifespan of the phagosomes.

Additionally, new HCs share many features with neighboring mature type II HCs, including myosin VIIA immunoreactivity, properly positioned nuclei, proper relative neck thicknesses, well defined stereocilia bundles, and basolateral processes. However, since we could not birth-date HCs in this study, we were not able to determine the degree to which new type II HCs are mature. Further, we did not perform any analyses that would inform on the functional status of the new HCs. Therefore, additional work is required to assess if HCs replaced in adult animals under normal conditions or after damage are functionally mature. These points were added to the Discussion (subsection “SCs transdifferentiate into type II HCs in normal mouse utricles”, fourth paragraph).

*Mechanistic insights: Conversion of supporting cells to hair cells appears to be a critical pathway for hair cell regeneration, however, the mechanistic details are lacking for this study to be sufficiently novel and of general importance. For example, what is the signal that triggers the supporting cell transdifferentiation? How does loss of supporting cells (by conversion to hair cells) affect vestibular function? How are the vestibular ganglion processes remodeled during hair cells loss and replacement?*

We agree that mechanisms of HC clearance and addition are very important to understand, and we are designing studies to address these questions. However, we agree with Essential Comment #7 that definition of mechanisms is beyond the scope of this initial report.

*Other issues:*

In the sixth paragraph of the subsection “Type I and II HCs are cleared by phagosomes from adult mouse utricles under normal conditions”: the authors report that in 5-10 week-old Swiss Webster mice, 71.7% of the on average 48 phagosomes per utricle are not hair cells, leaving the reader wondering what is the most intense phagosome activity in the utricle related to?

This is a great question. We suspect the phagosomes that are not associated with HCs are empty phagosomes that have already finished HC engulfment and HC markers such as myosin VIIa are no longer detectable by immunostaining. These empty phagosomes are likely lingering in the sensory epithelium before they are slowly resorbed as was described in aminoglycoside-treated adult mouse utricles by Monzack et al., 2015. To address your question, future live imaging studies are needed. We revised the Results (subsection “Type I and II HCs are cleared by phagosomes from adult mouse utricles under normal conditions”, fourth paragraph) to address this point.

*The paper shows SCs phagocytose both type I and II HCs, but only generate type II HC. Does the number of type I HCs decrease with age in the undamaged utricle?*

There are conflicting data on this point. In aged C57Bl/6NNia mice, Park et al. (1987) found a ~14% reduction in both type I and II HCs. However, Kirkegaard and Nyangaard (2005) found no evidence of HC loss in aged outbred mice for either HC type. We included these references and added more information for consequences of the possible loss of type I HCs in the Discussion (subsection “Significance of vestibular HC turnover in adult mice”, last paragraph).

*Although several morphological criteria are used to classify HCs as type I or type II in utricle whole-mounts, use of specific antibodies, as in Figure 1, would be necessary to make a compelling case.*

We opted to employ the 4 criteria that we have found, with considerable experience, to reliably define HC subtype. These criteria are thoroughly outlined in Materials and methods, illustrated in Video 4, published in our paper Pujol et al. (2014) and, for the case of some criteria, described clearly by several other investigators. The presence/absence of a basolateral process, and nuclear position, are sufficient to classify type I vs II HCs in most cases. In cases where we were not certain, we classified HCs as “unknown.”

Further, we were concerned that increasing the number of labels per utricle from 3 (DAPI, Myo7a, tdTomato) to 4 or 5 (DAPI, Myo7a, tdTomato, markers of type I and II HCs) would diminish the quality of immunolabeling and the reliability of analysis, while not providing much extra power in cellular identification.

*As shown by the authors, in Plp-CreERT2:ROSA26tdTomato mice, some Schwann cells in the stroma and cells in the region of transitional epithelium are also tdTomato-positive. Since F-actin enriched phagosomes are found in the SC nuclear layer, which is adjacent to the basal lamina and stroma, it could be a possibility that phagosomes are derived from Schwann cells. And phagosomes in the region near the transitional epithelium (as shown in Figure 2) may be derived from cells in the transitional epithelium. These possibilities should be considered in the Discussion.*

Please see our response above to Essential Comment #2.

*Figure 4: it would be helpful is the authors would explain what is the PCDH15-CD12 signal in the adult (Figure 4).*

Thank you for this suggestion. We clarified PCDH15-CD2 labeling in the revised figure legend text for this figure (now Figure 5).

*The last paragraph of the Discussion section does not flow with the rest of the text, seems misplaced.*

Great point. We rewrote the final 3 paragraphs of the Discussion. See Essential Comment #6.

*Reviewer #2:*

*This is an elegant paper reporting findings related to hair cell elimination and spontaneous regeneration in the vestibular epithelium.*

*"In mammals, vestibular HC production is reported to occur only during gestation and the first two postnatal weeks…"*

*This timing may be correct for mice but not necessarily all mammals. Rephrase please.*

Thank you. We made this correction (Introduction, second paragraph).

"Supporting cells (SCs) are glia-like cells". Also: "SCs, which resemble glia".

*SCs are epithelial cells. True, they have some functions and some molecular properties shared with glia, but the definition "glial-like" may not be accurate.*

Thanks for this suggestion. In a few places, we clarified that SCs are

epithelial in origin but have properties of glial cells (Introduction, third paragraph and subsection “Phagocytosis by SCs mediates HC clearance in normal mouse utricles”, first paragraph).

*Previous studies, including from the Seattle group, claimed that SCs are ejected from the sensory epithelium when eliminated by an insult. How do the current results conform to these findings? Please discuss to explain this.*

To the best of our knowledge there is no evidence that SCs are ejected from the epithelium. Perhaps this is a typo? After utricular HC damage by aminoglycosides, others have reported that the apical portion of the HC is cleaved and ejected but the cell body degenerates within the epithelium (Li et al., 1995; Bird et al., 2010; Monzack et al., 2015). To determine if the apical portion of HCs are also ejected under normal conditions, we acquired new data where normal utricles were labeled with antibodies to espin and PMCA2, which label stereocilia but not other structures in the sensory epithelium (Dumont et al., 2001; Li et al., 2004). While we detected bright labeling with each antibody in stereocilia, none of the HCs associated with phagosomes contained brightly labeled foci resembling stereocilia. This finding suggests that the apical part of the HC is ejected apically prior to translocation or that stereocilia degenerate during translocation, during normal clearance. Interestingly, espin antibodies did label the actin-rich spikes. Initially, we thought this could indicate the spike is derived from stereocilia. However, spikes were not labeled for antibodies to PMCA2, suggesting that this is not the case. We have added new images to Figure 2 (panels 2C-D”) and new text in the Results (subsection “Type I and II HCs are cleared by phagosomes from adult mouse utricles under normal conditions”, eighth paragraph) and Discussion (subsection “Phagocytosis by SCs mediates HC clearance in normal mouse utricles”, second paragraph).

*When SCs clear HCs after ototoxin-mediated injury, they first produce an actin cable to constrict the HC. This was shown by Meiteles in 1994 (PMID: 7528737). Add citation.*

Thank you for this reminder. We added this reference, as suggested.

*Assessment of the role of SCs in the death of HCs is presented in details as related to turnover, but not described for the DT lesion in the DTR mice, where it is even more important. Add data and discuss differences between turnover and response to trauma.*

This is an excellent point, which we addressed above, in Essential Comment #4.

*Figure 1 should indicate surface view more clearly. Could dispense with this image because all micrographs are at supporting cell level. A schematic at the level of SCs, would be more useful to add or replace.*

*Thank you for this comment. In the figure and the figure legend, we clarified that Figure 1 are surface (xy) views of the utricle. We opted to keep Figure 1, since they are intended as an introduction to the morphology of the utricle for readers not familiar with inner ear. Further, we feel Figure 1 is important, since we present micrographs in this orientation in several figures. However, given the frequency of the SC layer view, we modified Figure 1 to include a schematic of the SC nuclear layer as well.*

*Figure 1 legend: "In most panels…" should state here how the other panels are labeled.*

We edited the figure legend text to clarify the labeling of each panel (Figure 1 legend).

*The actin rod was first described by Sobin and later termed cytocaud by Kanzaki upon further characterization. Better citations would help. Cytocauds are compatible with survival of HCs for weeks or longer, yet they are clearly manifestations of stress and onset of degeneration. Therefore, their presence is not necessarily related to the mode of HC elimination.*

We added references to Sobin et al., 1982 and Kanzaki et al., 2002, as well as to Flock et al., 1979 when discussing cytocauds (subsection “Phagocytosis by SCs mediates HC clearance in normal mouse utricles”, second paragraph). The suggestion that the actin spike we observed (and compared to cytocauds) is the first sign that a HC is targeted for clearance is just a hypothesis proposed in our model. As stated in this same paragraph, further studies including live imaging are needed to test the model and determine whether the actin spike is a sign of HC degeneration or if it is unrelated. We clarified that this is our hypothesis and also presented the alternative that the actin spike does not play a role in HC clearance.

*Figure 1 is the merge? There seems to be more myosin at this level than in just the four areas attributed to ectopic HCs.*

Great point. The areas of myosin in Figure 1 not clearly associated with a HC nucleus are type II HC cytoplasmic processes. This has been revised in the figure legend, and a schematic was added to Figure 1, right panel.

*What is "brightest point projection image"?*

We changed this term to just “projection image” to prevent confusion.

*Addition of a neuronal marker for calyces would help reader to distinguish type I from type II HCs.*

As explained above (see our seventh response to reviewer 1), we used 4 robust criteria that we are confident can distinguish between type I and II HCs. When we were not sure, we defined cells as “unknown”.

*What is "basket-shape"? Baskets come in wide variety of shapes. A more specific geometric term should be used.*

Monzack et al. (2015) conducted live imaging of SC phagosome activity in aminoglycoside-treated adult mouse utricles and described the structure of the phagosomes as “basket-like.” The phagosomes we observed in normal adult mouse utricles resemble their findings and so we use the term “basket-like” to maintain consistency within the field.

To aid the readers in better understanding the term, we added a description of the shape the first time “basket-like” is used (subsection “Type I and II HCs are cleared by phagosomes from adult mouse utricles under normal conditions”, third and fourth paragraphs) and a new video file that rotates a 3D reconstruction of a basket-like phagosome with a spike piercing a HC (Video 1).

*It was not clear what the background of plp cre was, and whether Swiss Webster, CBA or c57b6 was the closest comparison.*

*Plp1-CreER^T2^* mice are on a mixed background. The background of all mouse strains used was updated in Materials and methods (subsection “Mouse models”).

*I would suggest not using the term "undamaged". An agent may not have been used to intentionally induce a lesion, but one cannot rule out the possibility that the cells targeted for phagocytosis are not damaged. It may be normal wear and tear but it is still damage.*

Great suggestion. We altered this terminology. See Essential Comment #3 above.

*In the second paragraph of the subsection “Immature HCs are present in undamaged utricles of adult mice”: larger format images similar to Figure 4 should be used to show the difference between ages in localization pcdh15 isoforms across utricular regions. This would be better support for the comparison to the distribution of atoh1.*

Unfortunately, lower magnification images of the PCDH15-CD2 labeling in a utricle are not helpful to show the distribution of HCs with short bundles, since individual bundles are obscured by long type I HC bundles and co-localization of the label with stereocilia is only defined clearly upon examination at high magnification. We addressed this question by constructing a map based of PCDH15-CD2-positive bundles using 100X images of the whole utricle (see new Figure 5).

*Figure 5 and Figure 6– aren't these addressing the same or at least overlapping questions? Type I HCs don't increase so they are not being generated by either proposed mechanism.*

*Data in Figure 5 (now Figure 6) were generated using Plp1-CreER^T2^:ROSA26^tdTomato^ mice, which labeled most SCs and allowed us to find HCs that were directly produced by tdTomato-labeled SCs at later time points after tamoxifen injection. Data in Figure 6 (now Figure 7) were generated with a different CreER line. Here we used Atoh1-CreER^TM^:ROSA26^tdTomato^ mice, which labeled ~ 40% of type II HCs, allowing us to address whether type II HCs convert into type I HCs over time. With experiments presented in Figure 5, we asked “do SCs give rise to HCs, and if so, which type?”. Experiments presented in Figure 6 were conducted to answer the question “do some type II HCs convert into type I HCs over time?”.*

*In the last paragraph of the subsection “Atoh1-CreER^TM^-labeled type II HCs did not convert into type I HCs under normal conditions”: Statistics on proportion is problematic because proportions are ratios, which are not normally distributed. They also do not address the potential interdependence of the numerator and denominator. A test for correlation between counts would be better – i.e. does slope differ between time points. Also, if the numerator and denominator are not truly dependent and independent, you might consider reduced major axis regression. It also seems like the data in Figure 5 might suffice to answer this.*

We deeply appreciated this comment and consulted with a statistician on the most appropriate test to analyze these data. After considerable deliberation, we selected an analysis of covariance (ANCOVA), since changes in labeled type I HCs is dependent on the number of total number of labeled cells per utricle. (In *Atoh1-CreER^TM^:ROSA26^tdTomato^*mice, we observed variability in the total number of tdTomato-positive cells across timepoints.) In addition, we determined that the best way to test our hypothesis (that type II HCs convert into type I HCs) was to determine whether the number of type I tdTomato-labeled HCs increased over time. A decrease in the number of type II tdTomato-labeled HCs does not necessarily mean conversion. These cells could have died or been phagocytosed. Analysis of tdTomato-labeled unknown HC counts is also not informative, since we were not able to determine their HC type. Therefore, we normalized the number of tdTomato-positive type I HCs to total labeled cells at each timepoint. By ANCOVA, there was no statistical difference between any of the timepoints, which suggests that type II HCs do not convert into type I HCs.

We replaced the graph in Figure 6 (now Figure 7) and changed the relevant text in Results (subsection “*Atoh1-CreER^TM^*-labeled type II HCs did not convert into type I HCs under normal conditions”, last paragraph). In addition, we added a new, later timepoint of 32 weeks post-tamoxifen, to allow us to assess conversion over a longer period.

*No need to include both sd and ci; one can be calculated from the other.*

eLife’s instructions request that both standard deviation and confidence intervals be reported in the manuscript. To improve the readability of the paper, most 95% confidence intervals were presented in Supplemental Tables (source data) rather than Results text. We created new Figure 5 and Figure 8 Supplemental Tables ([Supplementary-material SD3-data] and [Supplementary-material SD6-data]), so additional 95% confidence intervals could be removed from Results.

*Sample sizes are too small to justify a test assuming unequal variances; it may seem obvious that 11x difference in variances would be significant, but these sample sizes are too small for credible comparisons of variances, and it is not possible to know if they represent normal distributions.*

We re-analyzed the data in Figure 8 using an unpaired, two-tailed Student’s t-test and the difference between control and damaged samples remain significant. The text in the Results (subsection “Diphtheria toxin-induced HC ablation increased SC-to-type II HC transition in adult mouse utricles”, last paragraph) and Figure 8 Legend have been updated, and the use of a Welch’s correction was removed from the Materials and methods section.

*Reviewer #3:*

*[…] Specific comments:*

*1) The data presented suggest during normal ageing a small number of hair cells are removed from the epithelial surface and become associated with the actin-rich phagosomal structures described in the paper. It is not clear to me from the data presented whether the phagocytic process that is suggested in Figure 1 and Figure 2 is any different from the mechanisms that have been described and shown directly through live-imaging in both the avian and mammalian utricle. It seems very possible that the same process that has been described for ototoxically damage cells could be responsible for the removal of the cells described here. Is there any reason not to conclude that a similar process is used? This points warrants further discussion (note – see point 12 below)?*

This is a great point. We added new data for the analysis of phagosomes after HC damage in *Pou4f3^DTR^*mice and found some differences from our observations of phagocytosis in normal utricles (see Essential Comment #4). We also obtained new data to assess whether the apical portion of HCs are ejected from the epithelium in normal utricles by labeling with antibodies to the stereocilia proteins, espin and PMCA2 (see response to reviewer 2 comment 3).

*2) There is a description of "association" between phagosomes and hair cells e.g. subsection “Type I and II HCs are cleared by phagosomes from adult mouse utricles under normal conditions”, third paragraph, fourth and sixth paragraphs. A qualification of what constituted such "associations" and how there were determined by the observer is required. Does this association mean that there was clear evidence of engulfment of a hair cell within a phagosome?*

Thank you for making us aware of this lack in clarity. We added new text to better define the term “association” between phagosomes and HCs as: 1) a F-actin ring-like structure that fully encircled a HC body and was connected to a basket-like structure devoid of HC material or 2) a basket-like structure with one or more F-actin-rich processes that extended laterally and either contacted nearby HCs or pierced and entered their cytoplasm (subsection “Type I and II HCs are cleared by phagosomes from adult mouse utricles under normal conditions”, fourth paragraph). We also added a new video (Video 1), which shows a 3D reconstruction of a phagosome with a F-actin spike associated with a HC. The image can be rotated about the y-axis and shows the lattice structure of the basket-like phagosome.

*3) In order to call this structure a phagosome there needs to be better description of how the actin rich structures that are shown are linked to the supporting cells that presumably generated them – this is hard to see this from the images provided. Have the authors for example tried to reconstruct the phagosome structure in 3D from the confocal stacks they have?*

We used two mouse lines to label SCs (*Plp1-CreER^T2^:ROSA26^tdTomato^*and *Lfng-eGFP –* new to the revision) and found clear overlap of labeled SCs and F-actin phagosome structures (now Figure 4). As discussed above, we generated a movie of a 3D reconstruction of a basket-like phagosome with a F-actin spike piercing a HC in the SC layer (new Video 1) to help better visualize the phagosome structure. However, the mechanisms by which SCs produce F-actin phagosomes must be addressed in future experiments, using live imaging experiments or serial immuno-EM, as eluded to in the second paragraph of the subsection “Phagocytosis by SCs mediates HC clearance in normal mouse utricles” and in the second paragraph of the subsection “Significance of vestibular HC turnover in adult mice”.

*4) Is there something unusual about Swiss Webster mice? They have ~3x as many of the actin rich structures as the other two strains?*

See Essential comment #1 above.

*5) The phagosome-like structures are somewhat regionally restricted, being found predominantly in the peri-striolar region. There is little discussion of why this might be the case.*

This is a good point, which we addressed in the first paragraph of the subsection “Phagocytosis by SCs mediates HC clearance in normal mouse utricles”.

*6) Can the authors estimate the lifetime of the actin-rich phagosomal structures they describe (see note 12 below)?*

We attempted to estimate the lifespan of a phagosome in the second paragraph of the subsection “Significance of vestibular HC turnover in adult mice”, using live-cell imaging data from Monzack et al. (2015). This estimate was also used in this same section to estimate how many HCs are cleared per day and per week.

*7) How much actin is there in these structures compared to the actin-rich hair bundles of the normal hair cells? How about the spikes – the sequence showing the stack (Video 2) suggests very similar levels?*

We observed that the intensity of stereocilia labeling by F-actin is often much greater than the F-actin labeling in phagosomes and spikes. We postulate this is because phagosomes contain less actin or their location deeper in the epithelium decreases their exposure to the AlexaFluor-conjugated phalloidin dye compared to the stereocilia at the surface.

*8) How are the estimations for the% of Type I and Type II hair cells made (subsection “Atoh1-CreER^TM^-labeled type II HCs did not convert into type I HCs under normal conditions”, first paragraph)? I estimated 3% and 35% respectively using the numbers of tdTom labelled hair cells shown in the table for 7 weeks of age, a total hair cell number of 3600 and a ratio of 1.17:1.*

Thank you for pointing this out. We revised these estimates, using calculations below.

Assuming a ratio of type I:type II of 1.17:1 (Pujol et al., 2014), then 53.92% of HCs are type I and 46.08% of HCs are type II.

Assuming 3800 HCs per utricle (Golub et al., 2012),

53.92% = 2048.96, or 2049 type I HCs

46.08% = 1751.04, or 1751 type II HCs

For 1 week post-Tam in *Atoh1-CreER^TM^:ROSA26^tdTomato^*mice, we reported 50.3 type I HCs were labeled and 688.5 type II HCs were labeled.

50.3 labeled type I HCs/2049 type I HCs = 2.45%

688.5 labeled type II HCs/1751 type II HCs = 39.32%

We updated these numbers (2.5% and 39.3%) in the manuscript (subsection “*Atoh1-CreER^TM^*-labeled type II HCs did not convert into type I HCs under normal conditions”, first paragraph).

*9) At the end of the subsection “Diphtheria toxin-induced HC ablation increased SC-to-type II HC transition in adult mouse utricles” – it is stated that the rate of transdifferentiation is significantly increased. The data indicates that this is what happens but no rates are provided per se. Consider revising the sentence?*

Thank you for catching this. The term “rate” has been removed from the text, since the data were not generated with more than two timepoints.

*10) The finding that only Type II hair cells are replaced in the mammal has implications for the recovery of function since the Type I cells provide/send different information to brain. Does this finding concur with what is known/has been shown for the functional recovery that has been described in other work?*

This is a good discussion point that we missed. We added new text to the Discussion to address the functional consequences that may occur if type I HCs were progressively lost over time (subsection “Significance of vestibular HC turnover in adult mice”, last paragraph).

*11) The authors show in fate mapping experiments that ~2 hair cells are added per week. If we assume that the utricle maintains its normal number of cells over this period (which is thought to be the case) then the rate of removal should be the same. Even if the fate-mapping studies are not 100% effective and we assume a 75% efficacy then the simplest estimation is that there would be~3 (or just under) hair cells turned over per week, rather than the 10 suggested at the end of the subsection “Estimates of the rate of HCs turnover in adult mice”. Using this estimate we can see that the phagocytic structures that are described here is a somewhat long lived with lifetimes >1 week, indicating some differences to those phagosomes already described by others.*

We revised our estimation for the rate of HC turnover in the discussion and made conservative estimates solely based on the fate-mapping data since other analysis (phagosome, PDCH15-CD2, and ATOH1-GFP counts) lack a temporal perspective (subsection “Significance of vestibular HC turnover in adult mice”). In addition, other cells in the organ besides SCs (i.e. cells in the TE or stroma) could provide additional new HCs and were not fate-mapped in this study. We added this point as well.

*Other comments:i) Subsection “Type I and II HCs are cleared by phagosomes from adult mouse utricles under normal conditions”, fourth paragraph: How do the authors know that this is a calyceal afferent from the image shown in Panel 1F? This is quite hard to see and a marker would be required for definitive evidence. The tenascin data shown in Figure 1 provide some support for this although it is not entirely clear that the tenascin is within the actin-rich structure. Again – could some 3D reconstruction help?*

We can distinguish the calyx around type I HCs in plastic sections by its electron-lucent nature. Additionally, the calyx contains several small grayish electron-dense dots that are mitochondria. Figure 1 has been modified with additional arrows to point out these features, and the corresponding figure legend text now includes this information. We also modified the Results (subsection “Type I and II HCs are cleared by phagosomes from adult mouse utricles under normal conditions”, sixth paragraph).

We agree that it is challenging to see the exact localization of the tenascin labeling, but after looking at tenascin labeling in normal HCs, we are aware that any tenascin expression around a HC is indicative of type I status. Type II HCs have no tenascin immunolabeling in their vicinity.

*ii) Fig1L – it is hard to make out where the epithelial surface is perhaps this can be clarified.*

We made some edits to try to make this clearer. Figure 1 (now Figure 2) is a slice image taken at the level of the HC nuclei. Figure 1 (now Figure 2) are different views (yz and xz, respectively) of Figure 1 (now Figure 2) to help the reader see that the actin spike in the HC is below the apical surface of the epithelium. Additionally, Video 3 (was Video 2) is derived from this same field and should help the reader distinguish the location of the actin relative to the top of the epithelium. We edited Figure 2s legend to indicate that Video 3 is an excellent reference for Figure 2.

*iii) Figure 2 – clarify that this is a projection of the lower planes of the utricle – if it was a max projection of the full stack you would see mostly hair bundles in the image wouldn't you?*

This is now Figure 3. We added text to the figure legend to explain that the projection image was constructed to exclude the stereocilia, which would have obstructed the view of the phagosomes in the SC layer.

*iv) Figure 4 – is the edge of the sensory macula in the field of view or is it an oblique optical section in both cases?*

This is the edge of the macula, not an oblique view. We added white lines to mark the edge of the macula in Figure 4 and Figure 4 (now Figure 5) and revised the figure legend accordingly.

*v) I found Figure 5 hard to see due to the merging of the colours. It is clear that in C' there is a tdTom-positive and Myo7a-positive double labelled cell. Why does this cell have such a strong Myo7a signal – is this typical?*

This is now Figure 6. In our hands, variable immunofluorescent staining of myosin VIIa in HCs is typical in both utricles and cochlea. We added information to the figure legend pointing out that this particular HC happens to have increased staining compared to its neighbors, but is unlikely due to its being tdTomato-positive. We have also added new panels (Figure 6) showing myosin VIIa as a single channel.